

**The impact of multi-species surface chemical observations**
**assimilation on the air quality forecasts in China**
Zhen Peng[1*], Lili Lei[1], Zhiquan Liu[2*], Jianning Sun[1,3], Aijun Ding[1,3], Junmei Ban[2],
Dan Chen[4], Xingxia Kou[4], Kekuan Chu[1]
1 School of Atmospheric Sciences, Nanjing University, Nanjing, China
2 National Center for Atmospheric Research, Boulder, Colorado, USA
3 Institute for Climate and Global Change Research, Nanjing University, Nanjing,
China
4 Institute of Urban Meteorology, CMA, Beijing, China
**Abstract.** An Ensemble Kalman Filter data assimilation (DA) system has been
developed to improve air quality forecasts using surface measurements of $PM_{10}$, $PM_{2.5}$,
$SO_2$, $NO_2$, $O_3$ and CO together with an online regional chemical transport model, WRF-
Chem (Weather Research and Forecasting with Chemistry). This DA system was
applied to simultaneously adjust the chemical initial conditions (ICs) and emission
inputs of the species affecting $PM_{10}$, $PM_{2.5}$, $SO_2$, $NO_2$, $O_3$ and CO concentrations during
an extreme haze episode that occurred in early October 2014 over the East Asia.
Numerical experimental results indicate that ICs play key roles in $PM_{2.5}$, $PM_{10}$ and CO
forecasts during the severe haze episode over the North China Plain. The 72-h
verification forecasts with the optimized ICs and emissions performed very similarly to
the verification forecasts with only optimized ICs and the prescribed emissions. For the
first-day forecast, near perfect verification forecasts results were achieved. However,
with longer range forecasts, the DA impacts decayed quickly. For the $SO_2$ verification
forecasts, it was efficient to improve the $SO_2$ forecast via the joint adjustment of $SO_2$
ICs and emissions. Large improvements were achieved for $SO_2$ forecasts with both the
optimized ICs and emissions for the whole 72-h forecast range. Similar improvements
were achieved for $SO_2$ forecasts with optimized ICs only for just the first 3 h, and then
the impact of the ICs decayed quickly. For the $NO_2$ verification forecasts, both forecasts
performed much worse than the control run without DA. Plus, the 72-h $O_3$ verification





forecasts performed worse than the control run during the daytime, due to the worse
performance of the $NO_2$ forecasts, even though they performed better at night. However,
relatively favorable $NO_2$ and $O_3$ forecast results were achieved for the Yangtze River
delta and Pearl River delta regions.

**1 Introduction**

Predicting and simulating air quality remains a challenge in heavily polluted regions
(Wang et al., 2014; Ding et al. 2016). Chemical data assimilation (DA), which
combines observations and model simulations, is recognized as one effective method
to improve air quality forecasts. It has been widely used to assimilate aerosol
measurements from both ground-based and space-borne platforms, including surface
$PM_{10}$ observations (Jiang et al., 2013; Pagowski et al., 2014), surface $PM_{2.5}$
observations (Li et al., 2013; Zhang, 2016), Lidar observations (Yumimoto et al., 2007,
2008), aerosol optical depth products from AERONET (the AErosol RObotic
NETwork) (Schutgens et al., 2010a-b, 2012), and from various satellites (Sekiyama et
al., 2010; Liu et al., 2011; Dai et al., 2014). These studies indicate that assimilating
observations can substantially improve the spatiotemporal variations of aerosol in the
simulation and forecasts.
Aerosols are not only primarily emitted, but also with a larger portion secondary
formed through reactions with several gaseous-phases precursors and oxidants in the
atmosphere (Huang et al., 2014; Nie et al., 2014; Xie et al., 2015). So, observations of
trace gases are also useful in assimilating data for aerosol simulations and forecasts.
Efforts to assimilate atmospheric-composition observations, like $O_3$, $SO_2$, NO, $NO_2$,
CO, and $NH_3$, have also been made. For example, Elbern et al. (1997, 1999, 2000, 2001,
2007) developed a 4D-VAR (four-dimensional variational) system to assimilate surface
measurements of $O_3$, $SO_2$, NO and $NO_2$ to improve air quality forecasts with the joint
adjustment of initial conditions (ICs) and emission rates. Later, van Loon et al. (2000)
assimilated $O_3$ in the transport chemistry model LOTOS, based on an Ensemble Kalman
Filter (EnKF). Heemink and Segers (2002) attempted to reconstruct $NO_x$ and volatile
organic compound (VOC) emissions for $O_3$ forecasting by assimilating $O_3$. Carmichael



et al. (2003, 2008a, 2008b) developed 4D-VAR and EnKF systems to assimilate $O_3$ and
$NO_2$ to improve ICs and emission sources for $O_3$ forecasting. Hakami et al. (2005)
constrained black carbon (BC) emissions during the Asian Pacific Regional Aerosol
Characterization Experiment. Henze et al. (2007, 2009) estimated $SO_x$, $NO_x$ and $NH_3$
emissions based on a 4D-VAR method by assimilating surface sulfate and nitrate
aerosol observations. Other studies have estimated the $NO_x$ (van der et al., 2006, 2017;
Mijling et al., 2009, 2012, 2013; Ding. et al., 2015) and $SO_2$ emissions (van der et al.,
2017) based on an extended Kalman filter by assimilating $SO_2$ and $NO_2$ retrievals from
SCIAMACHY (SCanning Imaging Absorption spectroMeter for Atmospheric
CHartographY) and OMI (Ozone Monitoring Instrument). Barbu et al. (2009) applied
an EnKF to optimize the emissions and conversion rates using surface measurements
of $SO_2$ and sulfate. McLinden (2016) constrained $SO_2$ emissions using space-based
observations.

In recent years, severe haze pollution episodes have begun to occur more

frequently in China, especially in the megacity clusters of eastern China (e.g., Parrish
and Zhu, 2009; Sun et al., 2015; Zhang et al., 2015a). Thus, regional haze, especially
when accompanied by extremely high $PM_{2.5}$ concentrations, has drawn significant
research interest. However, there are large uncertainties involved in the numerical
prediction of atmospheric aerosols. During severe haze pollution episodes, air quality
models often underestimate the extreme peak mass concentration of particulate matter
(Wang et al., 2014). Previous studies have revealed that the assimilation of atmospheric-
composition observations can improve air quality forecasts by constraining the
uncertainties of both the chemical ICs and emissions (Tang et al., 2010, 2011, 2013,
2016; Miyazaki et al., 2012, 2013, 2014). Peng et al. (2017) demonstrated that
significant improvements in forecasting $PM_{2.5}$ can be achieved via the joint adjustment
of ICs and source emissions using an EnKF to assimilate surface $PM_{2.5}$ observations.

In 2013, China launched an atmospheric environmental monitoring system that

provides real-time and online atmospheric chemical observations, including $PM_{10}$,
$PM_{2.5}$, $SO_2$, $NO_2$, $O_3$, and CO (http://113.108.142.147:20035/emcpublish/). This
dataset provides an opportunity to improve air quality forecasts using DA. However,





such fruitful observations are less used in air quality forecast despite of large
discrepancy existed between the forecast and observations. But it is now possible to
estimate the impact on forecast improvement of simultaneously assimilating various
surface observations. Thus, we developed an EnKF system that can simultaneously
assimilate surface measurements of $PM_{10}$, $PM_{2.5}$, $SO_2$, $NO_2$, $O_3$ and CO to correct WRF-
Chem (Weather Research and Forecasting model with Chemistry) forecasts using the
Goddard Chemistry Aerosol Radiation and Transport (GOCART) aerosol scheme. As
an extension to Peng et al. (2017), the impact of simultaneously assimilating various
surface aerosol and chemical observations was investigated.
Sections 2 and 3 briefly describe the DA system and observations used in this
study, respectively. The experimental design is introduced in Section 4. Finally, the
assimilation results are presented in Section 5, before a brief summary in Section 6.

**2 DA system**
The DA system in this study was the same as the one used in Peng et al. (2017). It
can simultaneously analyze the chemical ICs and emissions with the assimilation of
surface $PM_{2.5}$ observations. A brief summary of the DA system is introduced here.
In every DA cycle, the ensemble emission scaling factors $\boldsymbol{\lambda}^f$ are first calculated
by the forecast model of scaling factors $\mathbf{M}_{SF}$ (see details of $\mathbf{M}_{SF}$ in section 2.2). Then,
the ensemble forecast emissions $\mathbf{E}^f$ are calculated using the following equation:
$$\mathbf{E}_{i,t} = \boldsymbol{\lambda}_{i,t}\mathbf{E}_t^p, (i = 1, \dots, N), \tag{1}$$
where $\mathbf{E}_t^p$ is the prescribed anthropogenic emission. The ensemble members of
chemical fields $\mathbf{C}^f$ are forecasted using WRF-Chem, forced by the forecast emissions
$\mathbf{E}^f$ whose ICs are previously analyzed concentration fields. Now, the background of
the joint vector, $\mathbf{x}^f = \left[\mathbf{C}^f, \boldsymbol{\lambda}^f\right]^T$, has been produced. Then, the analyzed state vector,
$\mathbf{x}^a = [\mathbf{C}^a, \boldsymbol{\lambda}^a]^T$, is optimized using an ensemble square root filter (EnSRF). Finally, the
assimilated emissions $\mathbf{E}^a$ can be obtained using equation (1).

**2.1 WRF-Chem model**




The model used to simulate the transport of aerosols and chemical species was the

WRF-Chem (Grell et al., 2005). As in Peng et al. (2017), we used version 3.6.1 and the

physical and chemical parameterization options are listed in Table 1. The model

computational domain covered almost the whole China and the horizontal resolution

was 40.5 km. Figure 1b shows our area of interest, the North China Plain (NCP). The

model included 57 vertical levels and the model top was 10 hPa.

The hourly prior anthropogenic emissions were based on the Multi-resolution

Emission Inventory for China (MEIC) (Li et al., 2014) for October 2010, instead of the

regional emission inventory in Asia (Zhang et al., 2009) for the year 2006 in Peng et al.

(2017). The reason we chose the MEIC-2010 was that the total emissions are reasonable

for cities over the NCP (Zheng et al., 2016). The original resolution of the MEIC-2010

is $0.25\,^\circ \times 0.25\,^\circ$, but has been processed to match the model resolution (40.5 km) (Chen

et al., 2016). No time variation was added to maintain objectivity in the prior

anthropogenic emissions.

**2.2 Forecast model of scaling factors**

In this work, the primary sources to be optimized were the emissions of $PM_{10}$, $PM_{2.5}$,

$SO_2$, NO, $NH_3$ and CO. The sources of $NH_3$ were analyzed because they also impact

greatly on the aerosols distribution. Thus, the emission scaling factors $\lambda_{i,t}^{f} =$

( $\lambda_{PM2.5}^{f}, \lambda_{PM10}^{f}, \lambda_{SO2}^{f}, \lambda_{NO}^{f}, \lambda_{NH3}^{f}, \lambda_{CO}^{f}$) were prepared by the forecast model of scaling

operator $\mathbf{M}_{SF}$ before WRF-Chem integration.

We used the same persistence forecast operator $\mathbf{M}_{SF}$ to forecast $\lambda_{i,t}^{f}$ as in Peng

et al. (2017). The forecast operator was developed by using the ensemble forecast

chemical fields. Thus,

$$\kappa_{i,t} = \frac{c_{i,t}^{f}}{\overline{\overline{c_t^{f}}}}, (i = 1, \dots, N), \tag{2}$$

$$(\kappa_{i,t})_{\text{inf}} = \beta\left(\kappa_{i,t} - \overline{\kappa_t}\right) + \overline{\kappa_t}, (i = 1, \dots, N), \tag{3}$$

$$\lambda_{i,t}^{p} = (\kappa_{i,t})_{\text{inf}}, \tag{4}$$

$$\lambda_{i,t}^{f} = \frac{1}{4}\left(\lambda_{i,t-3}^{a} + \lambda_{i,t-2}^{a} + \lambda_{i,t-1}^{a} + \lambda_{i,t}^{p}\right), (i = 1, \dots, N), \tag{5}$$





where $\mathbf{C}_{i,t}^{f}$ is the $i$th ensemble member of the chemical fields at time $t$, and $\overline{\mathbf{C}_{t}^{f}} =$
$\frac{1}{N}\sum_{i=1}^{N}\mathbf{C}_{i,t}^{f}$ is the ensemble mean; $\boldsymbol{\kappa}_{i,t}$ is the ensemble concentration ratios and $\overline{\boldsymbol{\kappa}_{t}}$ is
the ensemble mean of $\boldsymbol{\kappa}_{i,t}$ with values of 1; $\beta$ is the inflation factor to keep the
ensemble spreads of $\boldsymbol{\kappa}_{i,t}$ at a certain level; $\boldsymbol{\lambda}_{i,t-1}^{a}$, $\boldsymbol{\lambda}_{i,t-2}^{a}$ and $\boldsymbol{\lambda}_{i,t-3}^{a}$ are the previous
assimilated emission scaling factors.
In this study, the ensemble forecast chemical fields of $PM_{25}$, $PM_{10}$, $SO_2$, $NO$, $NH_3$
and CO of the previous assimilation cycle are respectively used to calculate the
emission scaling factors ( $\lambda_{PM2.5}^{f}, \lambda_{PM10}^{f}, \lambda_{SO2}^{f}, \lambda_{NO}^{f}, \lambda_{NH3}^{f}, \lambda_{CO}^{f}$). $\beta$ is chosen as 1.3,
1.4, 1.3, 1.2, 1.2, and 1.4 for the ensemble concentration ratios of $P_{25}$, $P_{10}$, $SO_2$, $NO$,
$NH_3$ and CO, respectively in Equation (3).
Then, the sources $\mathbf{E}_{i,t}^{f} = ( \mathbf{E}_{PM2.5}^{f}, \mathbf{E}_{PM10}^{f}, \mathbf{E}_{SO2}^{f}, \mathbf{E}_{NO}^{f}, \mathbf{E}_{NH3}^{f}, \mathbf{E}_{CO}^{f})$ are calculated
using equation (1).
From the perspective of $PM_{2.5}$ emissions, these include the unspeciated primary
sources of $PM_{2.5}$ $\mathbf{E}_{PM2.5}$, sulfate $\mathbf{E}_{SO4}$, nitrate $\mathbf{E}_{NO3}$, organic compounds $\mathbf{E}_{org}$ and
elemental compounds $\mathbf{E}_{BC}$. We updated $\mathbf{E}_{PM2.5}$, $\mathbf{E}_{SO4}$ and $\mathbf{E}_{NO3}$ (including the
nuclei and accumulation modes) following Peng et al. (2017).

**2.3 DA algorithm**
The assimilation algorithm employed was the EnSRF proposed by Whitaker and Hamill
(2002). The EnKF proposed by Evensen (1994) needs perturbations of observations in
practice. Compared to the original EnKF, the EnSRF obviates the need to perturb the
observations and avoids additional sampling errors introduced by perturbing
observations.
We used the same EnSRF as in Schwartz et al. (2012, 2014). The ensemble
member was chosen as 50. The localization radius was chosen as 607.5 km, so EnSRF
analysis increments were forced to zero at 607.5 km away from an observation (Gaspari
and Cohn, 1999). The posterior (after assimilation) multiplicative inflation factor was
chose as 1.2 for all the concentration analysis.




### 2.4 State variables


The DA system provides joint analysis of ICs and emissions following Peng et al.
(2017). Among them, 16 WRF-Chem/GOCART aerosol variables are included as the
state variables. Besides, chemical species, such as $SO_2$, $NO_2$ and $O_3$ are also included
because they are the most important gas-phase precursors or oxidants of the secondary
inorganic aerosols. CO is also assimilated because CO is an important tracer of
combustion sources, as well as a precursor of $O_3$ beyond $NO_2$ (Parrish et al., 1991). The
state variables of the emission scaling factors are $\boldsymbol{\lambda} =$
($\boldsymbol{\lambda}_{PM2.5}, \boldsymbol{\lambda}_{PM10}, \boldsymbol{\lambda}_{SO2}, \boldsymbol{\lambda}_{NO}, \boldsymbol{\lambda}_{NH3}, \boldsymbol{\lambda}_{CO}$).
Similar to weak-coupling DA, the DA system simultaneously updates both the ICs
and the emissions, but with no cross-variable update, in order to avoid the effects of
spurious multivariate correlations in the background error covariance that may develop
due to the limited ensemble size and errors in both the model and observations
(Miyazaki et al. 2012).
For the PM$_{2.5}$ observations, the observation operator is expressed as (Schwartz et
al., 2012)
$$y_{pm25}^{f} = \boldsymbol{\rho}_d[\mathbf{P_{25}} + 1.375\mathbf{S} + 1.8(\mathbf{OC_1} + \mathbf{OC_2}) + \mathbf{BC_1} + \mathbf{BC_2}$$

$$+\mathbf{D_1} + 0.286\mathbf{D_2} + \mathbf{S_1} + 0.942\mathbf{S_2}], \qquad (6)$$

where $\boldsymbol{\rho}_d$ is the dry air density; $P_{25}$ is the fine unspeciated aerosol contributions; S
represents sulfate; $OC_1$ and $OC_2$ are hydrophobic and hydrophilic organic carbon
respectively; $BC_1$ and $BC_2$ are hydrophobic and hydrophilic black carbon respectively;
$D_1$ and $D_2$ are dusts with effective radii of 0.5 and 1.4 μm espectively; $S_1$ and $S_2$ are
sea salts with effective radii of 0.3 and 1.0 μm espectively. In fact, PM$_{2.5}$ observations
were only used to analyze $P_{25}$, S, $OC_1$, $OC_2$ $BC_1$, $BC_2$, $D_1$, $D_2$, $S_1$, $S_2$ and $\boldsymbol{\lambda}_{PM2.5}$. Since
we had no $NH_3$ observations, PM$_{2.5}$ observations were also used to analyze $\boldsymbol{\lambda}_{NH3}$ (see
Table 2). For other control variables, PM$_{2.5}$ observations were not allowed to alter them.
For the PM$_{10}$ observations, the PM$_{10}$ observation operator is expressed as (Jiang
et al., 2013)



$$y^{\mathrm{f}}_{\mathrm{pm10}} = \boldsymbol{\rho}_{\mathrm{d}}[\mathbf{P_{10}} + \mathbf{P_{25}} + 1.375\boldsymbol{S} + 1.8(\mathbf{OC_1} + \mathbf{OC_2}) + \mathbf{BC_1} + \mathbf{BC_2}$$

$$+\mathbf{D_1} + 0.286\mathbf{D_2} + \mathbf{D_3} + 0.87\mathbf{D_4} + \mathbf{S_1} + 0.942\mathbf{S_2} + \mathbf{S_3}]. \quad (7)$$

Thus,
$$y^{\mathrm{f}}_{\mathrm{pm10-2.5}} = \boldsymbol{\rho}_{\mathrm{d}}[\mathbf{P_{10}} + \mathbf{D_3} + 0.87\mathbf{D_4} + \mathbf{S_3}], \qquad\qquad (8)$$

meaning that, in the assimilation experiments, we did not use the $PM_{10}$ observations
directly. In equation (13) and (14), $P_{10}$ denotes the coarse-mode unspeciated aerosol
contributions; $D_3$ and $D_4$ are dusts with effective radii of 2.4 and 4.5 µm respectively;
$S_3$ is sea salt with effective radii of 3.25 µm. We used the $PM_{10-2.5}$ observations (the
differences between the $PM_{10}$ observations and the $PM_{2.5}$ observations, $y^{\mathrm{o}}_{\mathrm{pm10-2.5}} =$
$y^{\mathrm{o}}_{\mathrm{pm10}} - y^{\mathrm{o}}_{\mathrm{pm10}}$ ) to analyze $P_{10}$, $D_3$, $D_4$, $S_3$ and $\boldsymbol{\lambda}_{\mathrm{PM10}}$ . In addition, $PM_{10-2.5}$
observations were used to analyze $D_5$ and $S_4$, since they are coarse-mode mineral dust
and sea salt aerosols. $PM_{10-2.5}$ observations were not allowed to impact other control
variables.
Moreover, as shown in Table 2, $SO_2$ observations were used to analyze the $SO_2$
concentration and $\boldsymbol{\lambda}_{\mathrm{SO2}}$ . $NO_2$ observations were used to estimate the NO, $NO_2$
concentration and $\boldsymbol{\lambda}_{\mathrm{NO}}$. CO observations were used to analyze the CO concentration
and $\boldsymbol{\lambda}_{\mathrm{CO}}$. And finally, $O_3$ observations were only used to analyze the $O_3$ concentration.

**3. Observations and errors**
The surface chemical observations used in this study were obtained from the Ministry
of Ecology and Environment of China. Altogether, there were 876 observational sites
over the model domain (Figure 1). At most sites, one measurement was selected
randomly for the assimilation experiment on a 0.1 °×0.1 °grid. Altogether, 355 stations
were kept for the model domain, where 133 assimilation stations were located on the
NCP and 40 stations were located in the Beijing–Tianjin–Hebei (BTH) region. Other
stations were used for verification purposes: 167 independent stations were located on
the NCP and 47 in the BTH region.
The observation error covariance matrix **R** included measurement errors and





representation errors. We assumed that **R** is a diagonal matrix (without observation
correlation).

Following Elbern et al. (2007), the measurement error $\varepsilon_0$ is defined as

$$\varepsilon_0 = a + b * \Pi_0, \qquad\qquad (9)$$

where $\Pi_0$ represents the measurements for PM$_{2.5}$, PM$_{10\text{-}2.5}$, SO$_2$, NO$_2$, CO or O$_3$ (units:
μg m$^{-3}$). A value of $a = 1.5$ and $b = 0.0075$ was chosen for PM$_{2.5}$, PM$_{10\text{-}2.5}$, SO$_2$,
and NO$_2$. For CO, $a = 10$ and $b = 0.0075$.

The representativeness error is defined as

$$\varepsilon_r = r\varepsilon_0\sqrt{\Delta x / \mathrm{L}}, \qquad\qquad (10)$$

where $r = 0.5$, $\Delta x = 40.5$ km (the model resolution), and $L = 3$ km due to the
lack of the information of the station type (Elbern et al., 2007).

Finally, the total error ($\varepsilon_t$) is defined as

$$\varepsilon_t = \sqrt{{\varepsilon_0}^2 + {\varepsilon_r}^2}, \qquad\qquad (11)$$

In order to ensure data reliability, the observations were subjected to quality

control before DA. Data values larger than a certain threshold were classified as
unrealistic and were not assimilated. The threshold values were chosen as 700, 800,
300, 300, 400 and 4000 μg m$^{-3}$ for PM$_{2.5}$, PM$_{10\text{-}2.5}$, SO$_2$, NO$_2$, O$_3$ and CO, respectively.
In addition, observations leading to innovations exceeding a certain value were also
omitted. These threshold values were chosen as 70 μg m$^{-3}$ for PM$_{2.5}$, PM$_{10\text{-}2.5}$, SO$_2$,
NO$_2$ and O$_3$. Also, 1500 μg m$^{-3}$ was chosen for CO.

**4. Experimental design**

The DA experiment followed that of Peng et al. (2017), in which the assimilation

of pure surface PM$_{2.5}$ measurements with the EnKF was performed to correct finer
aerosol variables and associated emissions. The experiment focused on an extreme haze
event that occurred in October 2014 over North China. The 50-member ensemble spin-
up forecasts were performed from 1 to 4 October 2014, in which the ICs, the lateral
boundary conditions and the emissions are perturbed by adding random noise. Then,
the observed PM$_{10}$, PM$_{2.5}$, SO$_2$, NO$_2$, O$_3$ and CO data starting from 5 to 16 October



were assimilated hourly to adjust the ICs and the corresponding emissions.
After that, two sets of 72-h forecasts were performed, each at 00:00 UTC from 6
to 15 October 2014, with hourly forecasting outputs for the assimilation experiment.
These two sets of forecasting experiments were conducted using the ensemble mean of
the concentration analysis as the ICs. One set of the experiments was forced by the
optimized emissions (denoted as fcICsEs), and the other was forced by the prescribed
anthropogenic emissions (denoted as fcICs). The aim was to use the difference between
the fcICsEs and fcICs to indicate the impact of the optimized emissions.
Moreover, we also run a control experiment. The ICs were based on the ensemble
mean of the spin-up forecasts at 00:00 UTC on 5 October 2014. The emissions were
the prescribed emissions.

**5. Results**
5.1 Ensemble performance
We begin by assessing the ensemble performance for the DA system. Figure 2 shows
the time series of the prior total spreads and the prior root-mean-square errors (RMSEs)
for $PM_{2.5}$, $PM_{10}$, and the four trace gases calculated against all observations in the BTH
region. It shows that the magnitudes of the total spreads were close to the RMSEs,
indicating that the DA system was well calibrated (Houtekamer et al., 2005).
Figure 3 shows the area-averaged time series extracted from the ensemble spread
of the six emission scaling factors ($\lambda^{f}_{PM2.5}$, $\lambda^{f}_{PM10}$, $\lambda^{f}_{SO2}$, $\lambda^{f}_{NO}$, $\lambda^{f}_{NH3}$ and $\lambda^{f}_{CO}$) in the
BTH region. It shows that the ensemble spread of all the scaling factors were very stable
throughout the ~10-day experiment period, which indicates that $\mathbf{M}_{SF}$ can generate
stable artificial data to generate the ensemble emissions. The value of the emission
scaling factors ranged from 0.2 to 0.6, indicating that the uncertainty of the assimilated
emissions was about 20%–60%.

5.2 Forecast improvements
In order to evaluate the overall performance of the DA system, time series of the hourly
pollutant concentrations from the control run, the analysis, and the first-day forecast of



the two forecasting experiments were compared with the independent observations in
the BTH region (Figure 4). Besides, model evaluation statistics (Table 3) were
calculated against independent observations from 6 to 16 October 2014. In addition,
biases and RMSEs were presented as a function of forecast range for the control,
analysis, and forecast experiments (Figures 5–7).

The control run did not perform very well, although it could capture the synoptic

variability and reproduce the overall pollutant levels when there was a severe haze event.
The statistics show that there were larger systematic biases and RMSEs and a smaller
correlation coefficient (CORR) for the control (see Table 3). The biases were $-34.1$,
$-77.7$, $-565.7$ and $-31$ $\mu g \cdot m^{-3}$ for $PM_{2.5}$, $PM_{10}$, CO, and $O_3$, respectively, from 6 to 16
October—about 29.7%, 44.5%, 42.9% and 53.9% lower than the corresponding
observed concentrations. During the severe haze episode from 8 to 10 October in
particular, when observed $PM_{2.5}$ were larger than 200 $\mu g \cdot m^{-3}$, the biases reached $-90.5$,
$-143.1$, $-911.8$ and $-39.1 \mu g \cdot m^{-3}$, respectively—about 44.4%, 51.9%, 49.2% and 55.7%
lower than the corresponding observed concentrations, suggesting a significant
systematic underestimation of the WRF-Chem simulation. Additionally, a significant
overestimation of 48.1 $\mu g \cdot m^{-3}$ was obtained for $SO_2$—about 145.8% higher than the
observed concentrations. As for the $NO_2$ simulation, WRF-Chem was able to
realistically describe the diurnal and synoptic evolution of $NO_2$ concentrations. The
model bias was 22.4 $\mu g \cdot m^{-3}$, which was about 39.7% higher than the observed $NO_2$.
These results were similar to the simulations of Chen et al. (2016). Most of the WRF-
Chem settings used here were the same as those used in Chen et al. (2016), except that
they used CBMZ (Carbon Bond Mechanism, version Z) and MOSAIC (Model for
Simulating Aerosol Interactions and Chemistry) as the gas-phase and aerosol chemical
mechanisms.

After the assimilation of surface observations, the time series of the hourly

pollutant concentrations from the analysis showed much better agreement with
observations than those from the control. The magnitudes of the bias and the RMSEs
decreased and the CORRs increased for all six species. The biases were 5.1, $-5.6$, 8.1,
$-8.3$, $-160.4$ and 2.1 $\mu g$ $m^{-3}$ for $PM_{2.5}$, $PM_{10}$, $SO_2$, $NO_2$, CO and $O_3$, respectively—



about 4.4%, −3.2%, 24.5%, −14.7%, −12.17% and 3.7% of the corresponding observed
concentrations, indicating that the analysis fields were very close to the observations.
The RMSEs were 51.5, 63.4, 27.9, 31.7, 618.9 and 31.1 µg m$^{-3}$, respectively—about
44.1%, 52.9%, 58.1%, 20.2%, 35.7% and 38.78% lower than the RMSEs of the control
run. The CORRs reached 0.891, 0.890, 0.540, 0.557, 0.705 and 0.753, respectively.
These statistics indicate that the DA system was able to adjust the chemical ICs
efficiently.

The PM$_{2.5}$, PM$_{10}$ and CO concentrations from both sets of forecasting experiments

benefitted substantially from the DA procedure, as expected. Smaller biases and
RMSEs were obtained for almost the entire 72-h forecast range (see Figures 5–7), as
compared with the control run. For the first-day forecast in particular, the model
performed almost perfectly. It faultlessly captured the diurnal and synoptic variability
of the pollutant (see figure 4), in a manner that was very close to that of the analysis.
The overall biases were 6.5, −11.9 and 100.4 µg m$^{-3}$ for PM$_{2.5}$, PM$_{10}$ and CO,
respectively; and the RMSEs were 77.8, 98.7 and 805.1 µg m$^{-3}$, respectively, in
fcICsEs24 (see Table 3). In fcICs24, the biases were 8.3, −10.3 and 130.2 µg m$^{-3}$,
respectively; and the RMSEs were 75.1, 95.9 and 838.2 µg m$^{-3}$, respectively (see Table
3). However, with longer-range forecasts, the impact of DA quickly decayed. The
relative reductions in RMSE mostly ranged from 30% to 5% for the second- and third-
day forecast. From the perspective of the impact of the assimilated emissions, fcICs
performed similarly to fcICsEs for PM$_{2.5}$, PM$_{10}$ and CO, indicating that ICs play key
roles in aerosol and CO forecasts during severe haze episodes, while the impact of
assimilated emissions seems negligible.

For the SO$_2$ verification forecast, however, fcICsEs performed much better than

both fcICs and the control run. Smaller biases and RMSEs were obtained for almost the
entire 72-h forecast range. At nighttime in particular (from 18 to 23 h, 42 to 47 h, and
66 to 73 h), when there was significant systematic overestimation in the control run,
both the biases and the RMSEs in fcICsEs were about 30% lower than those of the
control run. During the daytime (from 0 to 9 h, 24 to 33 h, and 48 to 57 h), fcICsEs still
performed slightly better, although the control run did a near perfect job. As for fcICs,



smaller biases and RMSEs were obtained for only the first 3 h. Then, the performance
was the same as the control run, indicating that the impact of the ICs had disappeared.
These results demonstrate the superiority of the assimilated emissions, and that the joint
adjustment of $SO_2$ ICs and emissions is an efficient way to improve the $SO_2$ forecast.
The $NO_2$ DA results for the independent sites showed really poor performance
(see Figures 5–7). Smaller biases were gained in the daytime of the experiment trials.
At nighttime, however, when the simulated $NO_2$ deviated considerably from the
observations in the control run, the biases of both sets of the validation forecasts became
even larger. Besides, almost all the RMSEs of both sets of the validation forecasts were
always larger than those of the control run.
The $O_3$ DA results were dependent on the $NO_2$ DA results in the daytime, due to
chemical transformation. Both the biases and the RMSEs were larger, as compared with
those of the control run (see Figures 5–7). However, at nighttime, when there was
significant systematic underestimation in the control run, the biases in fcICsEs had very
similar values to those of the analysis. Also, the biases in fcICs ranged between the
analysis and the control run; and the RMSEs of both sets of forecasting experiments
were about 10% smaller than those of the control run. All these results indicate that the
DA system performed well at night.

5.3 Emission optimization results
Besides improved pollutant forecasts, improved estimates of emissions were expected
from the joint DA procedure. The MEIC-2010 was constructed on the basis of annual
statistical books in which the data were often 2–3 years older than the actual year (Chen
et al., 2016). However, consistent efforts aimed at reducing and managing
anthropogenic emissions have been made over the past decade to mitigate air pollution.
Thus, there was a large difference between the emission year and our simulation year.
Besides, the spatial allocations of these emissions over small spatial scales, and the
monthly allocations, will also lead to some uncertainties. Lastly, the emissions
inventory cannot fully capture the day-to-day variability or the actual daily variations,
though its differentiation in terms of working days and weekend days, plus the daily





variations, can be taken into account in practical applications. However, in this
assimilation procedure, the differentiation in terms of working days and weekend days,
plus the daily variations, was ignored. Therefore, the prescribed anthropogenic
emissions were subject to large uncertainties.

Figures 8 and 9 display the spatial distribution of the prescribed emission rates and

the differences between the analysis and the prescribed emission rates of $PM_{2.5}$, $PM_{10}$,
$NH_3$, $SO_2$, NO and CO averaged over all hours from 6 to 16 October 2014 in the NCP
region. The assimilated emission rates of $PM_{2.5}$, $SO_2$, NO and CO were lower than the
prescribed emissions on the whole. In the BTH region especially, the differences
reached $-0.02$ $\mu g \cdot m^{-2} \cdot s^{-1}$, $-2.9$, $-8.8$ and $-24.65$ $mol \cdot km^{-2} \cdot hr^{-1}$, which was a reduction
of about 10%–20% of the prescribed emissions. For $PM_{10}$ emissions, the assimilated
values were very close to the prescribed ones, indicating that the prescribed $PM_{10}$
emissions had small uncertainties for the NCP region. For $NH_3$ emissions, the
assimilated values were a little larger than the prescribed emissions in large industrial
cities like Beijing, Tianjin, Baoding, Xingtai, Handan, and Taiyuan. However, they
were smaller than the prescribed emissions in agricultural regions, especially in
Shandong Province and Henan Province. However, in the BTH region, the assimilated
$NH_3$ emissions were very close to the prescribed emissions on the whole.

Figure 10 shows the time series of the emission scaling factors and the emissions.

As concluded in Peng et al. (2017), the forecast emission scaling factors changed with
the analyzed emission scaling factors due to the use of the time smoothing operator.
Besides, although the prescribed emissions were constant when designing the
assimilation experiment, the analyzed emission scaling factors showed obvious
variation with time, as did the analyzed emissions. For the assimilated $SO_2$ and NO
emissions in particular, the diurnal variations were perfect. In addition, the difference
between the assimilated emissions and the prescribed emissions were consistent with
those in Figures 8 and 9. The assimilated emissions of $PM_{2.5}$, $SO_2$, NO and CO were
apparently lower than the corresponding prescribed emissions. Whereas, the values of
the assimilated emissions of $PM_{10}$ and $NH_3$ were very close to their corresponding
prescribed emissions.




412 5.4 Discussion

413 From the results presented above, it is clear that improvements were achieved for

414 almost all the 72-h verification forecasts using the optimized ICs and emissions for

415 $PM_{2.5}$, $PM_{10}$, $SO_2$ and CO concentrations in the BTH region. However, the 72-h $NO_2$

416 verification forecasts performed much worse than the control run, due to the

417 assimilation. Plus, the 72-h $O_3$ verification forecasts performed worse than the control

418 run during the daytime, due to the worse performance of the $NO_2$ forecasts, although

419 they did perform better at night. However, relatively favorable $NO_2$ and $O_3$ forecast

420 results were gained for the Yangtze River delta and Pearl River delta (PRD) regions

421 (see Figure 11). In the PRD region, during the daytime, the three $NO_2$ forecasts (i.e.,

422 the control run, the fcICsEs, and the fcICs) performed similarly, and had relatively

423 small biases and RMSEs. At nighttime, when there was significant systematic

424 overestimation in the control run, the biases and the RMSEs in fcICsEs were much

425 smaller than those in the control run. For the $O_3$ 72-h verification forecasts, fcICsEs

426 performed much better than the control run, except for the first 8 h. Also, fcICs

427 improved the $O_3$ forecasts to some extent from the 9- to 72-h forecast range. These

428 results indicate that DA is still an effective way to improve $NO_2$ and $O_3$ forecasts.

429  Regarding the failure to improve the $NO_2$ and $O_3$ forecasts in the BTH region,

430 there are three likely factors. And certainly, $NO_2$ and $O_3$ forecasts in other areas are also

431 facing similar challenges.

432  Firstly, there are still some limitations for the EnKF method. EnKF assimilation is

433 influenced greatly by model errors and observation errors. For short-lived chemical

434 reactive species, such as $NO_2$ and $O_3$, they undergo highly complex nonlinear

435 photochemical reactions, even on timescales of hours, such that the forecast accuracy

436 is largely dependent on the chemical process as well as the physical transportation

437 process, the ICs, and the emissions. However, those complex photochemical reaction

438 processes are not precisely described in current chemical mechanisms, e.g.,

439 heterogeneous reactions (Yang et al., 2015), the photolysis of nitrous acid and $ClNO_2$

440 during daytime (Zhang et al., 2017), and so on. Therefore, on the one hand, there are



still large uncertainties for $NO_2$ and $O_3$ forecasts; whilst on the other hand, it is very
difficult for $NO_2$ and $O_3$ DA to accurately estimate the model errors with a limited
ensemble size. Thus, $NO_2$ and $O_3$ assimilations do not perform well (Elbern et al., 2007;
Tang et al., 2016). However, for $SO_2$ and CO, which are representative of long-lived
chemical reactive species, the chemical reaction process does not work
on timescales of hours, meaning that to some extent hourly chemical DA has the
potential to improve their forecasts. For CO in particular, due to its inertness, we might
be able to obtain high-quality ICs and emissions through DA. The primary sources of
aerosol are the dominant part of the atmospheric aerosol concentration. So, 72-h aerosol
forecasts may perform similarly to CO, albeit there are large uncertainties in the
chemical model.
Secondly, the analysis ICs and emissions are only a mathematical optimum under
the existing conditions. Only part of the chemical ICs and emissions are involved in the
DA experiment; and VOC ICs and emissions, which may greatly influence the $NO_2$ and
$O_3$ forecasts, were not included here because of the absence of VOC measurements.
Although we carried out two DA sensitivity experiments to adjust the VOC ICs and
emissions through the use of $NO_2$ or $O_3$ measurements, we were still unable to gain
improved $NO_2$ and $O_3$ forecasts in the BTH region in both DA experiments. VOC
measurements are needed to reduce uncertainties of VOC ICs and emissions. In
addition, almost all available data were observed in cities, and no observation stations
located in rural. Thus, the atmospheric environmental monitoring system was still
spatially heterogeneous.
Another important point is that there are still limitations to the current chemical
mechanisms used in our model, such as the treatment of model error. NO is the primary
species of $NO_x$ emissions in city areas, and reacts directly with $O_3$ to form $NO_2$ ($NO+O_3$
$\rightarrow NO_2+O_2$). Thus, $O_3$ concentrations may inversely correlate with $NO_2$ concentrations
at night. Consequently, air quality models may systematically underestimate $O_3$
concentrations. Currently, DA can only revise the ICs and the emissions in this work. It
cannot change the model performance, especially when there are certain uncertainties
for the meteorological simulation.





## 6. Summary

In this study, we developed an EnKF system to simultaneously assimilate surface measurements of $PM_{10}$, $PM_{2.5}$, $SO_2$, $NO_2$, $O_3$ and CO via the joint adjustment of ICs and source emissions. This system was applied to assimilate hourly pollution data while modeling an extreme haze event that occurred in early October 2014 over North China. In order to evaluate the impact of DA, two sets of 72-h verification forecasts were performed. One was conducted with the optimized ICs and emissions, and the other with only optimized ICs and the prescribed emissions. A control experiment without DA was also performed for comparison.

The results showed that both verification forecasts performed much better than the control simulations for $PM_{2.5}$, $PM_{10}$ and CO. Obvious improvements were achieved for almost the entire 72-h forecast range. For the first-day forecast especially, near perfect forecasts results were achieved. However, with longer-range forecasts, the impact of DA quickly decayed. In addition, the forecasts with only optimized ICs and the prescribed emissions performed similarly to that with the optimized ICs and emissions, indicating that ICs play key roles in $PM_{2.5}$, $PM_{10}$ and CO forecasts during severe haze episodes.

Also, large improvements were achieved for $SO_2$ forecasts with both the optimized ICs and emissions for the whole 72-h forecast range. However, similar improvements were achieved for $SO_2$ forecasts with the optimized ICs only for just the first 3 h, and then the impact of the ICs decayed quickly to zero. This demonstrates that the joint adjustment of $SO_2$ ICs and emissions is an efficient way to improve $SO_2$ forecasts.

Even though we failed to improve the $NO_2$ and $O_3$ forecasts in the BTH region, relatively favorable $NO_2$ and $O_3$ forecast results were gained in other areas. Also, the forecasts with both the optimized ICs and emissions performed much better than the forecasts with only optimized ICs and the prescribed emissions. These results indicate that there is still potential to improve $NO_2$ and $O_3$ forecasts via the joint adjustment of $SO_2$ ICs and emissions.



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

**List of Figures and Tables**





Figure 6. As in Figure 5 but for RMSE. Units: μg m$^{-3}$.

Figure 7. Normalized RMSE (assimilation divided by control) for fcICsEs and fcICs for PM$_{2.5}$, PM$_{10}$, SO$_2$ and CO.

Figure 8. Spatial distribution of the prescribed emissions (top panels) of PM$_{2.5}$ (left), PM$_{10}$ (middle), and NH$_3$ (right) and the corresponding time-averaged differences between the ensemble mean analysis and the prescribed values at the lowest model level averaged over all hours from 6 to 16 October 2014 in the NCP region. Units for PM$_{2.5}$ and PM$_{10}$ emissions: μg·m$^{-2}$·s$^{-1}$; and for NH$_3$ emissions: mol·km$^{-2}$·hr$^{-1}$.

Figure 9. As in Figure 8 but for SO$_2$ (left), NO (middle), and CO (right). Units for SO$_2$, NO and CO emissions: mol·km$^{-2}$·hr$^{-1}$.

Figure 10. Hourly area-averaged time series extracted from the analyzed emission scaling factors (black line), the forecast emission scaling factors (green dashed line), the analyzed emissions (blue line), and the prescribed emissions (blue dashed line) in the Beijing–Tianjin–Hebei region. Units for PM$_{2.5}$ and PM$_{10}$ emissions: μg·m$^{-2}$·s$^{-1}$; and for NH$_3$, SO$_2$, NO and CO emissions: mol·km$^{-2}$·hr$^{-1}$.




Table 1. WRF-Chem model configurations in this study.

| Parameterization | WRF-Chem Option |
|---|---|
| Aerosol scheme | Goddard Chemistry Aerosol Radiation and Transport (Chin et al., 2000, 2002) |
| Photolysis scheme | Fast-J (Wild et al., 2000) |
| Gas-phase chemistry | Regional Atmospheric Chemistry Mechanism (Stockwell et al., 1997) |
| Microphysics | the WRF single-moment 5 class scheme |
| Longwave radiation | Rapid Radiative Transfer Model longwave scheme (Mlawer et al., 1997) |
| shortwave radiation | Goddard shortwave radiation scheme (Chou and Suarez, 1994) |
| Planetary boundary layer | Yonsei University boundary layer scheme (Hong et al., 2006) |
| cumulus parameterization | Grell-3D scheme |
| Land-surface model | NOAH (Chen and Dudhia, 2001) |
| Dust and sea salt emissions | Goddard Chemistry Aerosol Radiation and Transport (Chin et al., 2002) |






Table 2. State vectors in the data assimilation system.

| Observations | $PM_{2.5}$ | $PM_{10-2.5}$ | $SO_2$ | $NO_2$ | CO | $O_3$ |
|---|---|---|---|---|---|---|
| Mass concentration | $P_{25}$, S, $OC_1$, $OC_2$ $BC_1$, $BC_2$, $D_1$, $D_2$, $S_1$, $S_2$ | $P_{10}$, $D_3$, $D_4$, $D_5$ $S_3$, $S_4$, | $SO_2$ | NO, $NO_2$ | CO | $O_3$ |
| Scaling factors | $\lambda_{PM2.5}$, $\lambda_{NH3}$ | $\lambda_{PM10}$ | $\lambda_{SO2}$ | $\lambda_{NO}$ | $\lambda_{CO}$ | — |






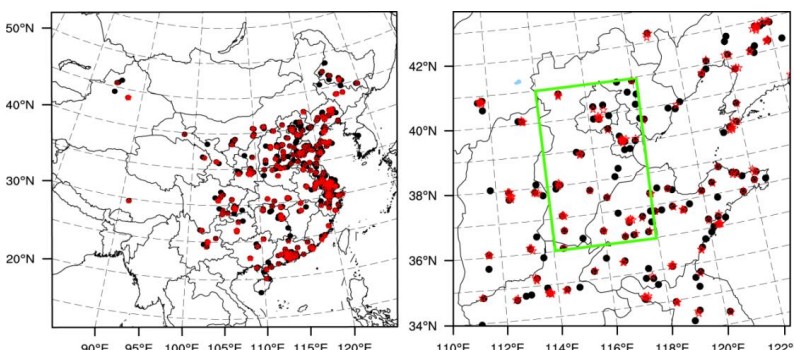


Figure 1. The model domain (left) and the North China Plain (right). Black dots are
the observational sites used for assimilation, and red stars are the observational sites
used for validation. The green frame marks the Beijing–Tianjin–Hebei region.





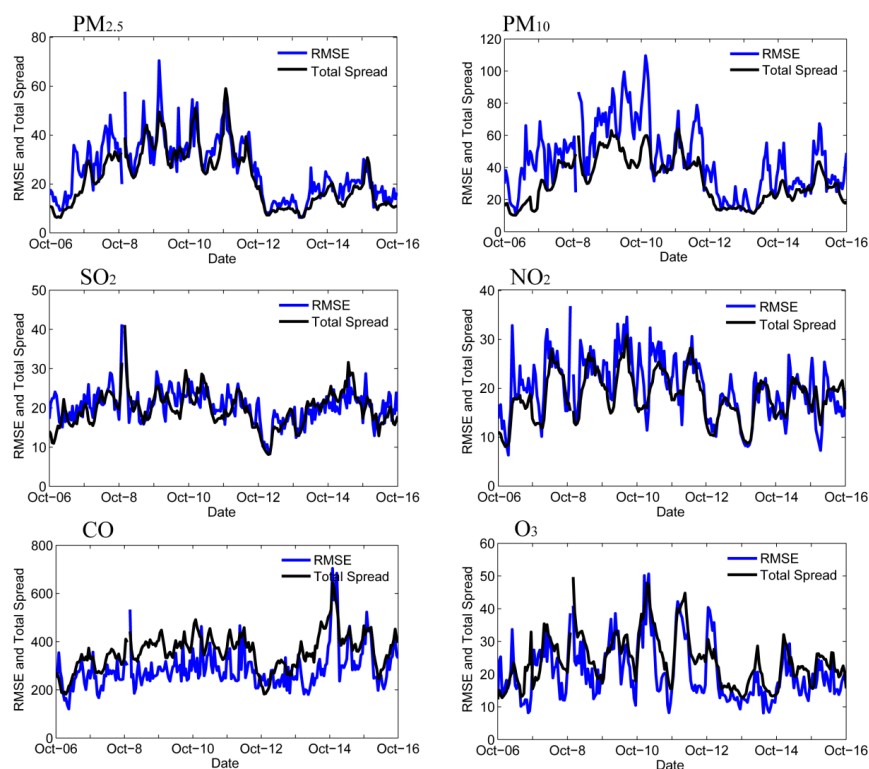


Figure 2. Time series of prior ensemble mean RMSE (blue line) and total spread
(black line) for $PM_{2.5}$, $PM_{10}$, $SO_2$, $NO_2$, CO and $O_3$ concentrations aggregated over all
observations over the Beijing–Tianjin–Hebei region. Units for all these variables are

$\mu g\ m^{-3}$.






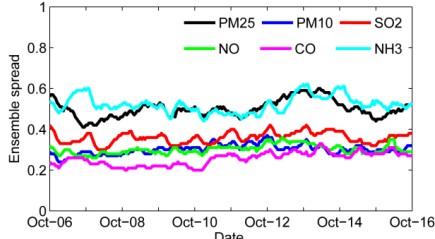


Figure 3. Time series of the area-averaged ensemble spread for the emission scaling
factors over the Beijing–Tianjin–Hebei region.




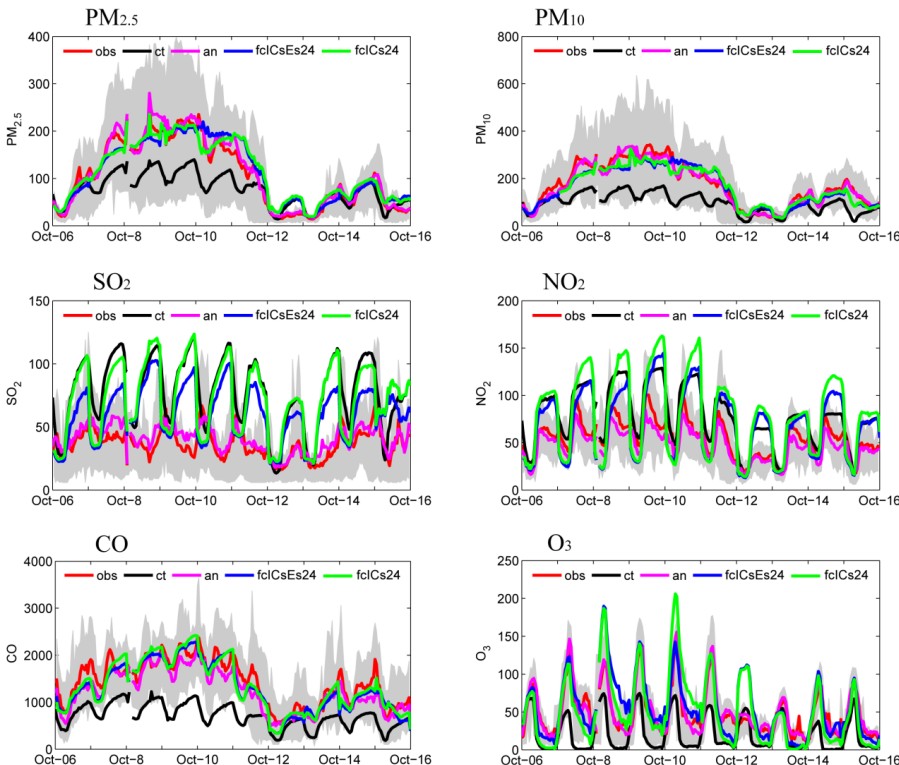


Figure 4. Time series of the hourly pollutant concentrations in the Beijing–Tianjin–
Hebei (BTH) region obtained from observations (red line), the control run (black
line), the analysis (pink line), the first-day forecast from fcICsEs (fcICsEs24, blue
line), and the first-day forecast from fcICs (fcICs24, blue line). The observations were
obtained from the 47 independent sites in the BTH region. The modelled time series
were interpolated to the 47 independent sites using the spatial bilinear interpolator
method. The shaded backgrounds indicate the distribution of the observations, where
the top edge represented the 90[th] percentile and the bottom edge the 10[th] percentile.
Units: $\mu g \ m^{-3}$.





Table 3. Comparison with observations of the surface PM$_{2.5}$ mass concentrations in the Beijing–
Tianjin–Hebei region from the control experiment, the assimilation experiment, and the first-day
forecast, over all analysis times from 6 to 16 October 2014. Units: μg m$^{-3}$.

| Species | Experiment | Mean observed value | Mean simulated value | BIAS | RMSE | CORR |
|---|---|---|---|---|---|---|
| PM$_{2.5}$ | Control | | 80.7 | −34.1 | 92.1 | 0.740 |
| | Analysis | 114.8 | 119.9 | 5.1 | 51.5 | 0.891 |
| | fcICsEs24 | | 121.2 | 6.5 | 77.8 | 0.736 |
| | fcICs24 | | 123.1 | 8.3 | 75.1 | 0.748 |
| PM$_{10}$ | Control | | 96.9 | −77.7 | 134.6 | 0.691 |
| | Analysis | 174.6 | 169.0 | −5.6 | 63.4 | 0.890 |
| | fcICsEs24 | | 162.7 | −11.9 | 98.7 | 0.716 |
| | fcICs24 | | 164.3 | −10.3 | 95.9 | 0.726 |
| SO$_2$ | Control | | 81.1 | 48.1 | 66.6 | 0.088 |
| | Analysis | 33.0 | 41.1 | 8.1 | 27.9 | 0.540 |
| | fcICsEs24 | | 62.0 | 29.0 | 51.2 | 0.120 |
| | fcICs24 | | 75.7 | 42.7 | 65.8 | 0.038 |
| NO$_2$ | Control | | 78.8 | 22.4 | 39.7 | 0.545 |
| | Analysis | 56.4 | 48.0 | −8.3 | 31.7 | 0.557 |
| | fcICsEs24 | | 71.8 | 15.4 | 46.2 | 0.408 |
| | fcICs24 | | 82.8 | 26.4 | 55.5 | 0.414 |
| CO | Control | | 752.3 | −565.7 | 962.7 | 0.354 |
| | Analysis | 1318.0 | 1157.5 | −160.4 | 618.9 | 0.705 |
| | fcICsEs24 | | 1418.4 | 100.4 | 805.1 | 0.476 |
| | fcICs24 | | 1448.2 | 130.2 | 838.2 | 0.439 |
| O$_3$ | Control | | 26.5 | −31.0 | 50.8 | 0.463 |
| | Analysis | 57.5 | 59.6 | 2.1 | 31.1 | 0.753 |
| | fcICsEs24 | | 63.5 | 6.0 | 49.0 | 0.460 |
| | fcICs24 | | 58.98 | 1.5 | 50.5 | 0.478 |







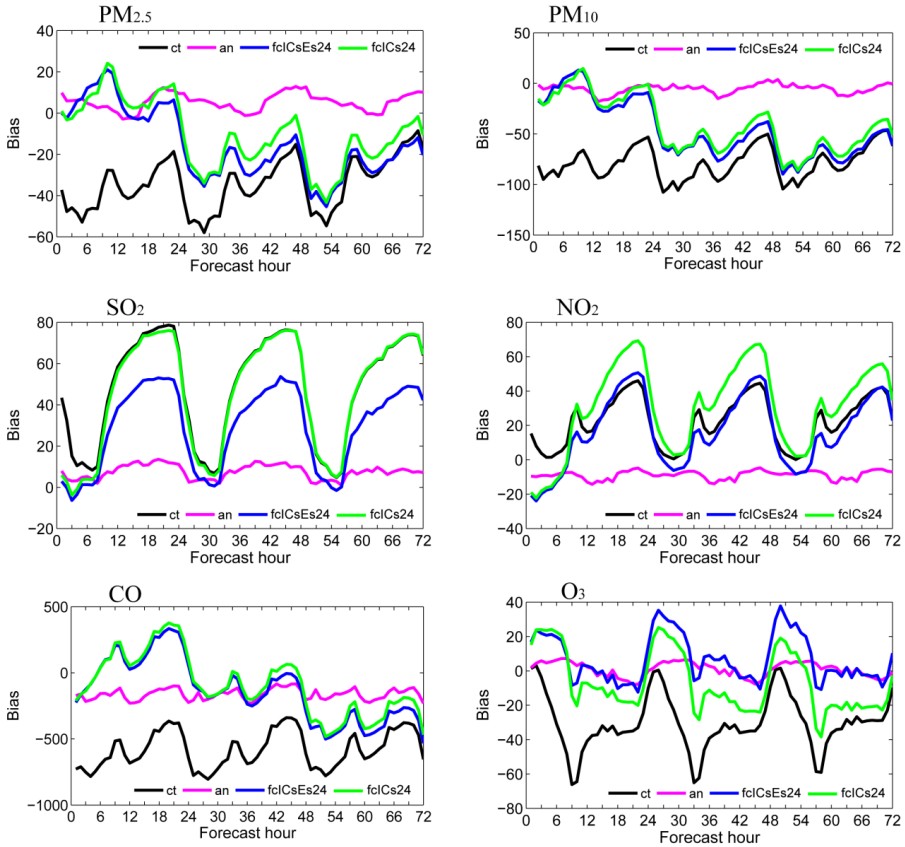

Figure 5. Bias of surface PM$_{2.5}$, PM$_{10}$, SO$_2$, NO$_2$, CO and O$_3$ as a function of forecast range calculated against all the independent observations over the Beijing–Tianjin–Hebei region shown in Figure 1. The 72-h forecasts were performed at each 0000 UTC from 6 to 14 October 2014 and the statistics were computed from 6 to 14 October. Units: μg m$^{-3}$.






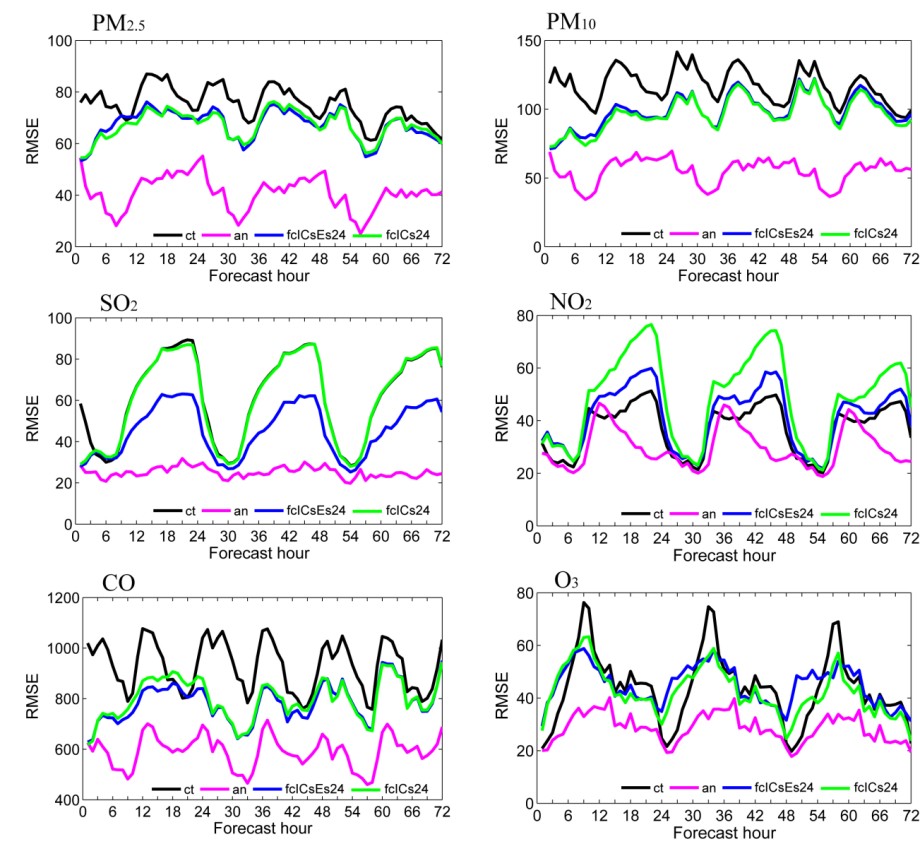


Figure 6. As in Figure 5 but for RMSE. Units: $\mu g\ m^{-3}$.




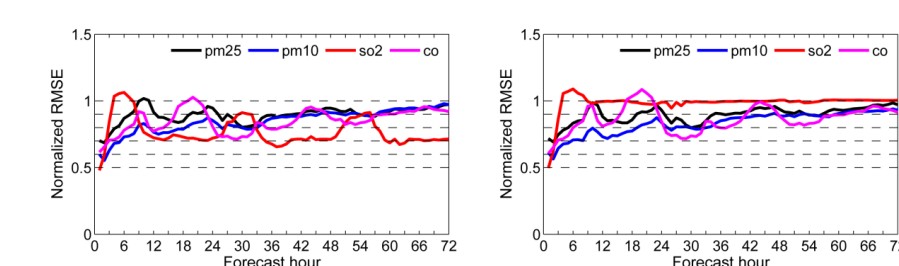


Figure 7. Normalized RMSE (assimilation divided by control) for fcICsEs and fcICs
for $PM_{2.5}$, $PM_{10}$, $SO_2$ and CO.




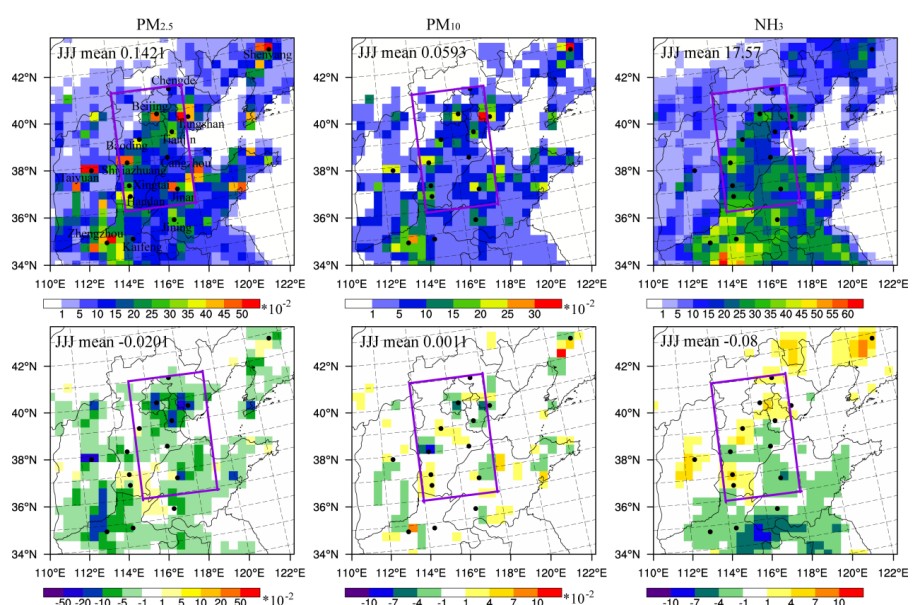


Figure 8. Spatial distribution of the prescribed emissions (top panels) of PM$_{2.5}$ (left), PM$_{10}$ (middle), and NH$_3$ (right) and the corresponding time-averaged differences between the ensemble mean analysis and the prescribed values at the lowest model level averaged over all hours from 6 to 16 October 2014 in the NCP region. Units for PM$_{2.5}$ and PM$_{10}$ emissions: μg·m$^{-2}$·s$^{-1}$; and for NH$_3$ emissions: mol·km$^{-2}$·hr$^{-1}$.







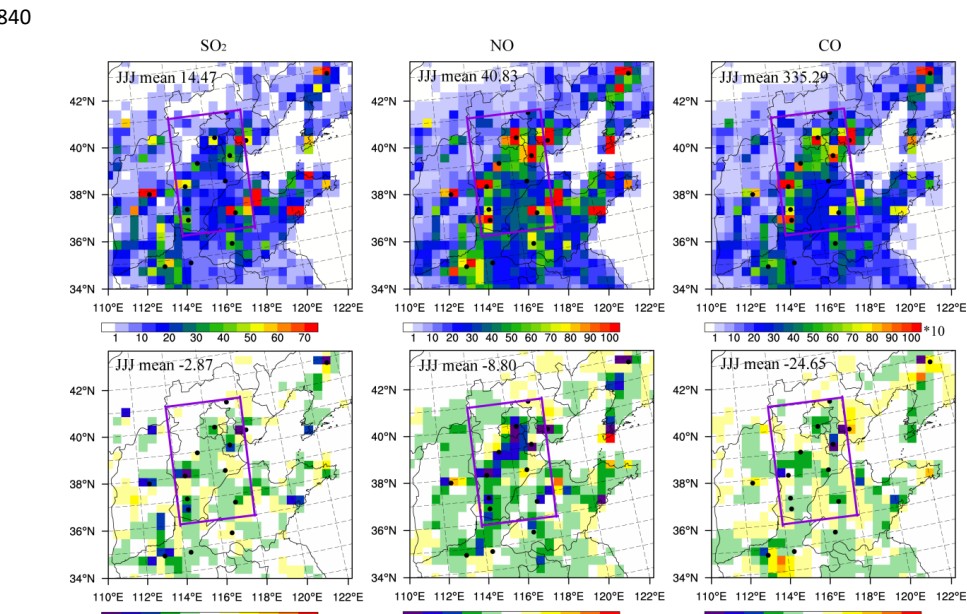


Figure 9. As in Figure 8 but for SO$_2$ (left), NO (middle), and CO (right). Units for SO$_2$, NO
and CO emissions: mol·km$^{-2}$·hr$^{-1}$.




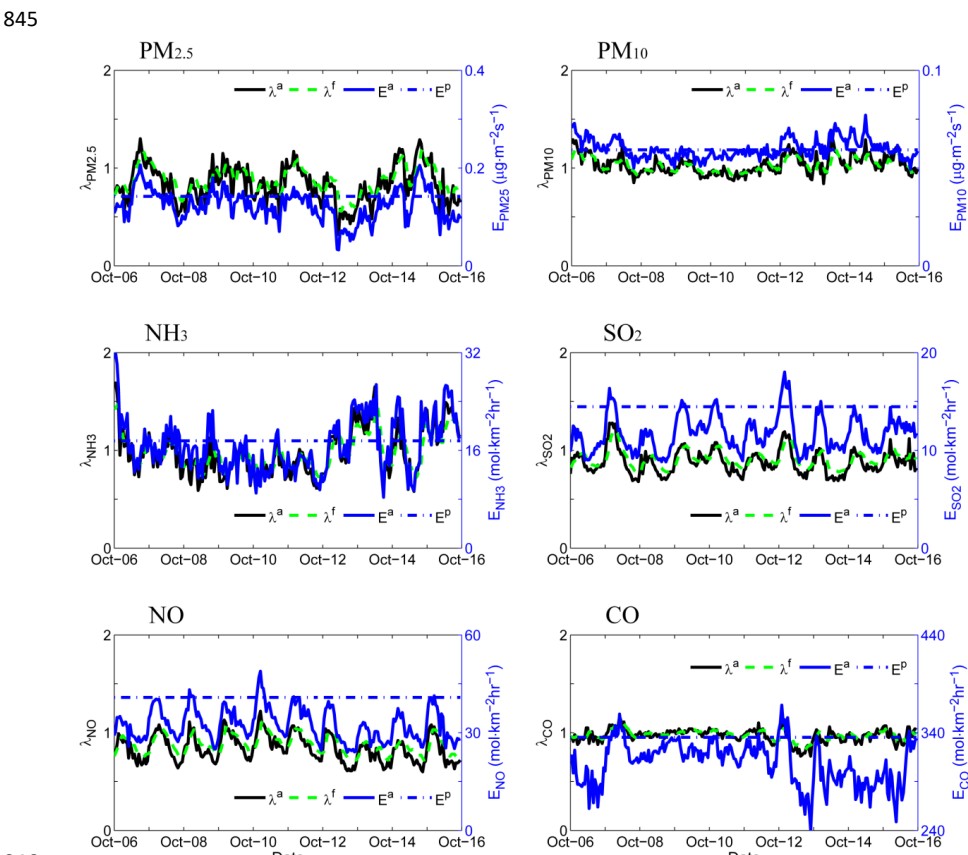


Figure 10. Hourly area-averaged time series extracted from the analyzed emission
scaling factors (black line), the forecast emission scaling factors (green dashed line),
the analyzed emissions (blue line), and the prescribed emissions (blue dashed line) in
the Beijing–Tianjin–Hebei region. Units for PM$_{2.5}$ and PM$_{10}$ emissions: $\mu g \cdot m^{-2} \cdot s^{-1}$;
and for NH$_3$, SO$_2$, NO and CO emissions: $mol \cdot km^{-2} \cdot hr^{-1}$.






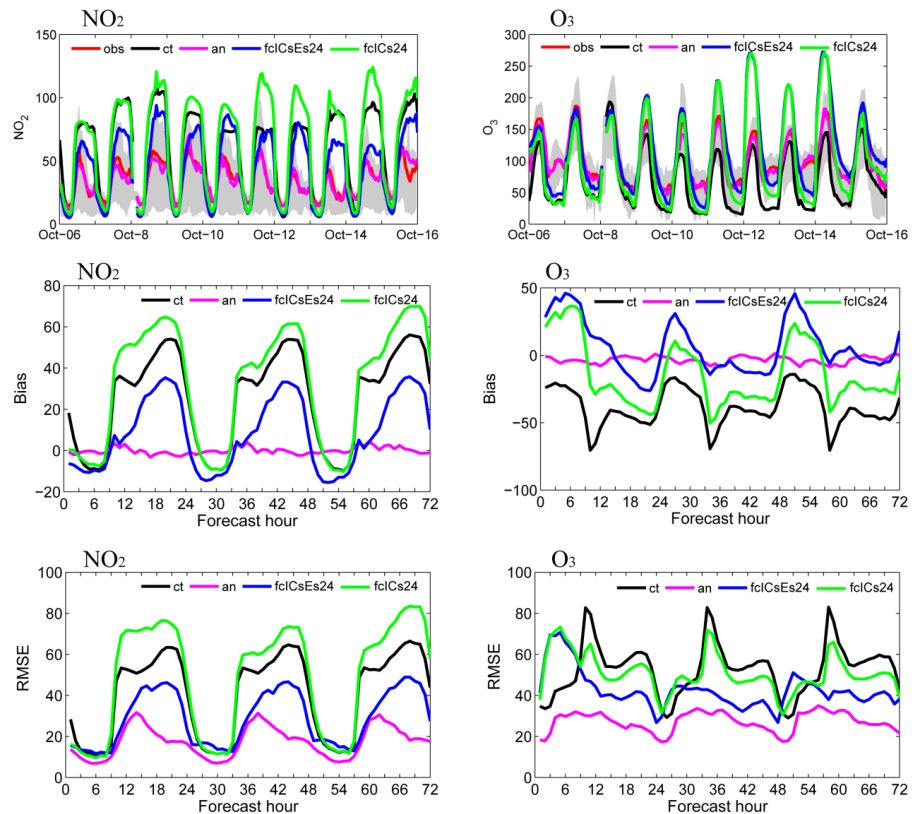


Figure 11. NO$_2$ and O$_3$ time series of the hourly pollutant concentrations in the Pearl
River Delta region (PRD, 21 °–24 °N, 112.5 °–115 °E) obtained from observations (red
line), the control run (black line), the analysis (pink line), the first-day forecast from
fcICsEs (fcICsEs24, blue line), and the first-day forecast from fcICs (fcICs24, blue
line). The bias and RMSEs of surface NO$_2$ and O$_3$ as a function of forecast range
calculated against all the independent observations (34 sites) over the PRD region.
Units: μg m$^{-3}$.