# Peer review of "The impact of multi-species surface chemical observations assimilation on the air quality forecasts in China"

_Atmospheric Chemistry and Physics, 2018_

## Referee Comment (RC1) · Anonymous Referee #1 · 4 Sep 2018

Summary and general comments:

This manuscript investigated the application of ensemble Kalman filter (EnKF) for constraining the atmospheric chemical species including PM10, PM2.5, SO2, NO2, O3 and CO. The simultaneous assimilation of various surface air quality measurements improved the representation of the initial conditions and emission factors of aforementioned species, as well as their 72-hours forecasts. This investigation on the assimilation of various air quality observations for a severe haze pollution event provides a promising case study for the regional air-quality modeling. I would recommend the minor revision with the considerations of several issues as listed below.

[Figure]

List of minor comments:

1) Section 2.1: Which dataset (reanalysis) did you use for the meteorological initial and boundary conditions? Were the perturbations also added to the meteorology? If not, please add one or two sentences to mention that the uncertainty of the meteorology forecasts is not considered in this study.

2) L107-108: Are emission scaling factors $\lambda$ spatially varying?

3) L154-156: Why the inflation factors for the chemical species $\beta$ are different among the variables? Could you please provide the strategy you took to find these values?

4) L257-259: How did you perturb the initial conditions, lateral boundary conditions and emissions? In other words, please provide how you estimated the background uncertainty and spatial correlations (i.e. background covariance structures) for the chemical state variables in adding perturbations?

5) L275-279 and Figure 2: This is very promising. I would imagine that the impacts of other sources of uncertainties in air-quality forecast that were not directly considered in this study (such as chemical schemes and parameterizations in forecast model, and meteorology) were indirectly considered through the well-calibrated inflations of state variables. Could you please make a comment about the impacts of these other uncertainty sources in discussion section? I believe it would be helpful for the future readers of this manuscript.

6) Figure 4: It is not clear to me what "The shaded backgrounds indicate the distribution of the observations, where the top edge represented the 90th percentile and the bottom edge the 10th percentile" means. Does this distribution represent the observation values of individual sites in the Beijing–Tianjin–Hebei (BTH) region? Are other (red, black, pink, blue and light green) thick lines average of all sites in BTH region? The purpose to show these two values together is unclear to me, since the grey shaded line and other thick lines do not seem to be comparable each other. I would recommend to

add more explanations about this figure, or to remove the grey shaded lines.

List of specific comments:

1) L174: Please change "chose" to "chosen".

2) L296: I think "was able to" better fits with this context than "could".

3) Figure 4: The acronyms of "an" and "ct" is not described (although they can be guessed from the figure caption). Could you please add the explanation of those acronyms in the figure captions, such as "the analysis (referred to as "an", pink line)"?

4) Figure 11: Please add the explanation of grey shaded lines in the top panels.

---

## Referee Comment (RC2) · Anonymous Referee #2 · 4 Sep 2018

There is not much to criticize about the manuscripts as it relies on the assimilation methodology previously described by Peng et al. (2017). Since the assimilation experiment was conducted over a ten-day period it is uncertain if the conclusions about different performance of forecasts for various species would hold in a general.

The most interesting are results on emission factors. Did you encounter negative lambdas and if so what did you do about them? An ultimate test of the optimized emissions would compare a simulation using the optimized emissions with a control. Would an ENFK run with concentrations as state vectors using optimized emissions be identical to the EnKF run with concentrations and emission factors as the state vectors?

[Figure]

Link http://113.108.142.147:20035/emcpublish (p. 3) would be a valuable data source on pollution over China for many users but the access requires installation of Microsoft Silverlight a software for watching videos. That seems odd and is not be allowed on government computers. Could that be ameliorated?
* * *

---

## Author Comment (AC1) · 31 Oct 2018

**Response to Reviewer #1's comments:**

We thank Referee # 1 for his thoughtful comments and suggestions that have helped to improve our manuscript. Our responses to comments (in bold style) and the corresponding changes to the manuscript are detailed below.

**Summary and general comments:**

**This manuscript investigated the application of ensemble Kalman filter (EnKF) for constraining the atmospheric chemical species including $PM_{10}$, $PM_{2.5}$, $SO_2$, $NO_2$, $O_3$ and CO. The simultaneous assimilation of various surface air quality measurements improved the representation of the initial conditions and emission factors of aforementioned species, as well as their 72-hours forecasts. This investigation on the assimilation of various air quality observations for a severe haze pollution event provides a promising case study for the regional air-quality modeling. I would recommend the minor revision with the considerations of several issues as listed below.**

**List of minor comments:**

**1) Section 2.1: Which dataset (reanalysis) did you use for the meteorological initial and boundary conditions? Were the perturbations also added to the meteorology? If not, please add one or two sentences to mention that the uncertainty of the meteorology forecasts is not considered in this study.**

The meteorological initial and boundary conditions were provided by the National Centers for Environmental Prediction Global Forecast System (GFS). The temperature, water vapor, velocity, geopotential height and dry surface pressure fields of the meteorological initial and boundary conditions were perturbed by adding Gaussian random noise with a zero mean and static background error covariances (Torn et al., 2006) to generated the 50 ensemble members by WRFDA. We have added these sentences in Line 274-278, Page 10.

**2) L107-108: Are emission scaling factors $\lambda$ spatially varying?**

Yes, the emission scaling factors $\boldsymbol{\lambda}$ here are spatially varying. In our system, we use the ensemble forecast chemical fields $\mathbf{C}_{i,t}^{f}$ and the previous DA cycles' analysis scaling factors $\boldsymbol{\lambda}_{i,t-3}^{a}$, $\boldsymbol{\lambda}_{i,t-2}^{a}$, $\boldsymbol{\lambda}_{i,t-1}^{a}$ to evaluate the emission scaling factors $\boldsymbol{\lambda}_{i,t}^{f}$. Since $\mathbf{C}_{i,t}^{f}$ were spatially varying, the ensemble concentration ratios $\boldsymbol{\kappa}_{i,t} = \mathbf{C}_{i,t}^{f}/\overline{\mathbf{C}_{t}^{f}}$ were spatially varying too. Thus, $\boldsymbol{\lambda}_{i,t}^{f} = \frac{1}{4}\left(\boldsymbol{\lambda}_{i,t-3}^{a} + \boldsymbol{\lambda}_{i,t-2}^{a} + \boldsymbol{\lambda}_{i,t-1}^{a} + \boldsymbol{\lambda}_{i,t}^{p}\right) = \frac{1}{4}\left(\boldsymbol{\lambda}_{i,t-3}^{a} + \boldsymbol{\lambda}_{i,t-2}^{a} + \boldsymbol{\lambda}_{i,t-1}^{a} + (\boldsymbol{\kappa}_{i,t})_{\text{inf}}\right) = \frac{1}{4}\left(\boldsymbol{\lambda}_{i,t-3}^{a} + \boldsymbol{\lambda}_{i,t-2}^{a} + \boldsymbol{\lambda}_{i,t-1}^{a} + \beta\left(\boldsymbol{\kappa}_{i,t} - \overline{\boldsymbol{\kappa}_{t}}\right) + \overline{\boldsymbol{\kappa}_{t}}\right)$ were spatially varying.

We have added these sentences in Line 156-158, Page6.

**3) L154-156: Why the inflation factors for the chemical species $\beta$ are different among the variables? Could you please provide the strategy you took to find these values?**

Peng et al. (2015) first used the forecast model of scaling operator $\mathbf{M}_{\text{SF}}$ to prepare the ensemble emission scaling factors $\boldsymbol{\lambda}^{f}$ in order to optimize all $CO_2$ fluxes as a whole at grid scale. In Peng et al. (2015), the ensemble spread of $\boldsymbol{\kappa}_{i,t} = \mathbf{C}_{i,t}^{f}/\overline{\mathbf{C}_{t}^{f}}$ was very small (ranging from 0 to 0.08 in most area at model-level 1), though the values of the ensemble spread of $\mathbf{C}_{i,t}^{f}$ after inflation could reach 1 to 14 ppmv in most area at model-level 1. Therefore, covariance inflation was used to keep it at a certain level. After covariance inflation, the ensemble spread of $\lambda_{i,t}^{a}$ ranged from 0.1 to 0.8 in most model area for $\beta = 70$. Besides, several sensitive experiments were performed to investigate $\beta$ (10, 50, 60, 70, 75, 80, 100). The ensemble spread of $\lambda_{i,t}^{a}$ ranged from 0.05 to 1.25 for $\beta$ =60, 70, 75, 80. And the $CO_2$ DA system worked comparatively well for $\beta$ =60, 70, 75, 80. The assimilated CO2 fluxes deviated markedly from the "true" CO2 fluxes when the ensemble spread of $\lambda_{i,t}^{a}$ were too small for $\beta$ =10, 50 or when the ensemble spread of $\lambda_{i,t}^{a}$ were too large for $\beta$ =100. Though $CO_2$ fluxes inversion was another topic, we mentioned it here because this experience was very helpful for us to develop the joint DA system for aerosol.

In Peng et al. (2017), four emission scaling factors, $\lambda^f_{PM2.5}$, $\lambda^f_{SO2}$, $\lambda^f_{NO}$ and $\lambda^f_{NH3}$, are optimized in Peng et al. (2017) when the pure surface PM2.5 observations are assimilated. We use the same inflation factor $\beta$ to keep the ensemble spreads of $\lambda^f_{PM2.5}$, $\lambda^f_{SO2}$, $\lambda^f_{NO}$ and $\lambda^f_{NH3}$ at a certain level. Several sensitive experiments were performed to investigate $\beta$ (1.2, 1.5, 1.8, 2, 2.5). It is seemed that reasonable results can be obtained when the ensemble spread of the emission scaling factors $\lambda^f_{PM2.5}$ ranged from 0.1 to 1. Finally, $\beta = 1.5$ was chosen in Peng et al. (2017). The area-averaged ensemble spreads of $\lambda^f_{PM2.5}$, $\lambda^f_{SO2}$, $\lambda^f_{NO}$ and $\lambda^f_{NH3}$ were stably distributed around 0.5, 1.0, 1.5 and 0.8 respectively over the three sub-regions: Beijing–Tianjin–Hebei region, Yangtze River delta and Pearl River delta. It is apparent that the ensemble spread of $\lambda^f_{SO2}$ and $\lambda^f_{NO}$ is a little large due to the same $\beta$.

Therefore, it is better to choose different inflation factors for different emission scaling factors. We have performed several sensitive experiments to determine the value of $\beta$ over a 2-day period before the experiments written in the manuscript. The criterion we choose $\beta$ is to keep the ensemble spread of the scaling factors ranging from 0.1 to 1 in most model area. Finally, $\beta$ is chosen as 1.3, 1.4, 1.3, 1.2, 1.2, and 1.4 for $\lambda^f_{PM2.5}$, $\lambda^f_{PM10}$, $\lambda^f_{SO2}$, $\lambda^f_{NO}$, $\lambda^f_{NH3}$ and $\lambda^f_{CO}$ (See ReFig. 1)

Perhaps there are very few negative values for $(\kappa_{i,t})_{inf}$ after inflation. A quality control procedure is performed for $(\kappa_{i,t})_{inf}$ before further appliance. All these negative data were set as 0 in this work. Then $(\kappa_{i,t})_{inf}$ were re-centered to ensure the ensemble mean values of $(\kappa_{i,t})_{inf}$ were all 1. Then, another quality control procedure is performed for $\lambda^a_{i,t}$ to keep them positive. Thus, all $\lambda^f_{i,t}$ and $\lambda^a_{i,t}$ could be positive.

We have added these sentences in Line 158-163, 166-169, Page 6.

[Figure]

[Figure]

ReFig. 1. Spatial distribution of the ensemble spread for $\lambda^f_{PM2.5}$, $\lambda^f_{PM10}$, $\lambda^f_{SO2}$, $\lambda^f_{NO}$, $\lambda^f_{NH3}$ and $\lambda^f_{CO}$ at the lowest model level at 0000 UTC 6 October 2014 in the NCP region.

4) **L257-259: How did you perturb the initial conditions, lateral boundary conditions and emissions? In other words, please provide how you estimated the background uncertainty and spatial correlations (i.e. background covariance structures) for the chemical state variables in adding perturbations?**

Before the first DA cycle, a 50-member ensemble of four-day spin-up forecasts was performed, with perturbed meteorological initial conditions (ICs), lateral boundary conditions (LBCs) and emissions, from 0000 UTC 1 October to 2300 UTC 4 October 2014. The perturbed meteorological ICs and LBCs are created by adding Gaussian random noise (Torn et al., 2006) to the temperature, water vapor, velocity, geopotential height and dry surface pressure fields of the products of the National Centers for Environmental Prediction Global Forecast System (GFS) by WRFDA. The perturbed emissions were generated also by adding Gaussian random noise with a standard deviation of 10 percent of the corresponding anthropogenic emissions. The aerosol ICs were zero and the aerosol LBCs were idealized profiles embedded within the WRF/Chem model. They are the same as in Peng et al. (2017). It is noted that the perturbed emissions were only used in the initial part.

In the DA part, the ICs were the analysis of the previous DA cycle, the meteorological LBCs were the perturbed LBCs. The anthropogenic emissions, $E^f_{PM2.5}$, $E^f_{PM10}$, $E^f_{SO2}$, $E^f_{NO}$, $E^f_{NH3}$, $E^f_{CO}$, sulfate $E^f_{SO4}$ and nitrate $E^f_{NO3}$ are calculated by using the forecast emission scaling factors. Other species, such as the organic compounds $E_{org}$ and elemental compounds $E_{BC}$, are perturbed by adding Gaussian

random noise. Since the emissions are calculated by EQ. (1), their background uncertainties and the spatial correlations are completely dependent on those of the corresponding emission factors. The forecast scaling factors are calculated by EQ. (2) ~ (5). And no other perturbations are added to the scaling factors; no other correlations are assumed for the scaling factors.

The experimental design is the same as in Peng et al (2017). We have rewritten briefly in Section 4 to avoid the repetition (Line 272-293, Page 10-11).

**5) L275-279 and Figure 2: This is very promising. I would imagine that the impacts of other sources of uncertainties in air-quality forecast that were not directly considered in this study (such as chemical schemes and parameterizations in forecast model, and meteorology) were indirectly considered through the well-calibrated inflations of state variables. Could you please make a comment about the impacts of these other uncertainty sources in discussion section? I believe it would be helpful for the future readers of this manuscript.**

It is true that the impacts of other sources of uncertainties in air-quality forecast (such as chemical schemes and parameterizations in forecast model, and meteorology) were not directly considered through the well-calibrated inflations of state variables. EnKF assimilation is influenced greatly by model errors and observation errors. But it is very difficult to accurately evaluate the uncertainties of models, though the covariance inflation technique was simply applied for all state variables to roughly compensate for model errors. Therefore, we can only obtain suboptimal results through EnKF assimilation.

We have added the above paragraph in Lines 476-482, Page 17.

**6) Figure 4: It is not clear to me what "The shaded backgrounds indicate the distribution of the observations, where the top edge represented the 90th percentile and the bottom edge the 10th percentile" means. Does this distribution represent the observation values of individual sites in the Beijing–Tianjin–Hebei (BTH) region? Are other (red, black, pink, blue and light green) thick lines**

**average of all sites in BTH region? The purpose to show these two values together is unclear to me, since the grey shaded line and other thick lines do not seem to be comparable each other. I would recommend to add more explanations about this figure, or to remove the grey shaded lines.**

Yes. the grey shaded line represent the distribution of the observation values of individual sites in the Beijing–Tianjin–Hebei (BTH) region. Other (red, black, pink, blue and light green) thick lines represent the average values of all sites in BTH region. No more information could be obtained from the grey shaded line since the average values of observations (red line) were shown. Thus we remove the grey shaded lined in Figure 4.

**List of specific comments:**

**1) L174: Please change "chose" to "chosen".**

We have revised the word in Line 188, Page 6.

**2) L296: I think "was able to" better fits with this context than "could".**

We have changed the word in Line 328, Page 12.

**3) Figure 4: The acronyms of "an" and "ct" is not described (although they can be guessed from the figure caption). Could you please add the explanation of those acronyms in the figure captions, such as "the analysis (referred to as "an", pink line)"?**

We have changed theese in Line 848-855, Page 32.

**4) Figure 11: Please add the explanation of grey shaded lines in the top panels.**

We remove the grey shaded lined in Figure 11, similar to Figure 4.

---

## Author Response (AR1)

Oct. 31, 2018.

*Atmos. Chem. Phys.*

RE: Manuscript Number: acp-2018-768

Dear Editors:

Thank you very much for your kind decision letter on our paper entitled "The impact of multi-species surface chemical observations assimilation on the air quality forecasts in China" (acp-2018-768). We are grateful for the helpful comments from you and the reviewers. We have changed the manuscript according to the reviewer's suggestions. The main changes include: 1) A simulation using the optimized emissions from 5 to 16 October 2014 were performed to investigate the impact of optimized emissions on chemical simulations; 2) We have rewritten the experimental design in Section 4. All the scientific questions have been resolved in the revised version (Please see details in it). So we hope this manuscript will be published in ACP. We are looking forward to hearing from you soon.

Sincerely Yours,

Zhen Peng

**Response to Reviewer #1's comments:**

We thank Referee # 1 for his thoughtful comments and suggestions that have helped to improve our manuscript. Our responses to comments (in bold style) and the corresponding changes to the manuscript are detailed below.

**Summary and general comments:**

**This manuscript investigated the application of ensemble Kalman filter (EnKF) for constraining the atmospheric chemical species including $PM_{10}$, $PM_{2.5}$, $SO_2$, $NO_2$, $O_3$ and CO. The simultaneous assimilation of various surface air quality measurements improved the representation of the initial conditions and emission factors of aforementioned species, as well as their 72-hours forecasts. This investigation on the assimilation of various air quality observations for a severe haze pollution event provides a promising case study for the regional air-quality modeling. I would recommend the minor revision with the considerations of several issues as listed below.**

**List of minor comments:**

**1)    Section 2.1: Which dataset (reanalysis) did you use for the meteorological initial and boundary conditions? Were the perturbations also added to the meteorology? If not, please add one or two sentences to mention that the uncertainty of the meteorology forecasts is not considered in this study.**

The meteorological initial and boundary conditions were provided by the National Centers for Environmental Prediction Global Forecast System (GFS). The temperature, water vapor, velocity, geopotential height and dry surface pressure fields of the meteorological initial and boundary conditions were perturbed by adding Gaussian random noise with a zero mean and static background error covariances (Torn et al., 2006) to generated the 50 ensemble members by WRFDA. We have added these sentences in Line 274-278, Page 10.

    **2) L107-108: Are emission scaling factors $\lambda$ spatially varying?**

Yes, the emission scaling factors $\lambda$ here are spatially varying. In our system, we use the ensemble forecast chemical fields $\mathbf{C}_{i,t}^{f}$ and the previous DA cycles' analysis scaling factors $\lambda_{i,t-3}^{a}$, $\lambda_{i,t-2}^{a}$, $\lambda_{i,t-1}^{a}$ to evaluate the emission scaling factors $\lambda_{i,t}^{f}$. Since $\mathbf{C}_{i,t}^{f}$ were spatially varying, the ensemble concentration ratios $\kappa_{i,t} = \mathbf{C}_{i,t}^{f}/\overline{\mathbf{C}_{t}^{f}}$ were spatially varying too. Thus, $\lambda_{i,t}^{f} = \frac{1}{4}\left(\lambda_{i,t-3}^{a} + \lambda_{i,t-2}^{a} + \lambda_{i,t-1}^{a} + \lambda_{i,t}^{p}\right) =$ $\frac{1}{4}\left(\lambda_{i,t-3}^{a} + \lambda_{i,t-2}^{a} + \lambda_{i,t-1}^{a} + (\kappa_{i,t})_{inf}\right) = \frac{1}{4}\left(\lambda_{i,t-3}^{a} + \lambda_{i,t-2}^{a} + \lambda_{i,t-1}^{a} + \beta\left(\kappa_{i,t} - \overline{\kappa_{t}}\right) + \overline{\kappa_{t}}\right)$ were spatially varying.

We have added these sentences in Line 156-158, Page6.

**3) L154-156: Why the inflation factors for the chemical species $\beta$ are different among the variables? Could you please provide the strategy you took to find these values?**

Peng et al. (2015) first used the forecast model of scaling operator $\mathbf{M}_{SF}$ to prepare the ensemble emission scaling factors $\lambda^{f}$ in order to optimize all $CO_2$ fluxes as a whole at grid scale. In Peng et al. (2015), the ensemble spread of $\kappa_{i,t} = \mathbf{C}_{i,t}^{f}/\overline{\mathbf{C}_{t}^{f}}$ was very small (ranging from 0 to 0.08 in most area at model-level 1), though the values of the ensemble spread of $\mathbf{C}_{i,t}^{f}$ after inflation could reach 1 to 14 ppmv in most area at model-level 1. Therefore, covariance inflation was used to keep it at a certain level. After covariance inflation, the ensemble spread of $\lambda_{i,t}^{a}$ ranged from 0.1 to 0.8 in most model area for $\beta = 70$. Besides, several sensitive experiments were performed to investigate $\beta$ (10, 50, 60, 70, 75, 80, 100). The ensemble spread of $\lambda_{i,t}^{a}$ ranged from 0.05 to 1.25 for $\beta$ =60, 70, 75, 80. And the $CO_2$ DA system worked comparatively well for $\beta$ =60, 70, 75, 80. The assimilated CO2 fluxes deviated markedly from the "true" CO2 fluxes when the ensemble spread of $\lambda_{i,t}^{a}$ were too small for $\beta$ =10, 50 or when the ensemble spread of $\lambda_{i,t}^{a}$ were too large for $\beta$ =100. Though $CO_2$ fluxes inversion was another topic, we mentioned it here because this experience was very helpful for us to develop the joint DA system for aerosol.

In Peng et al. (2017), four emission scaling factors, $\lambda_{PM2.5}^f$, $\lambda_{SO2}^f$, $\lambda_{NO}^f$ and

$\lambda_{NH3}^f$, are optimized in Peng et al. (2017) when the pure surface PM2.5 observations are assimilated. We use the same inflation factor $\beta$ to keep the ensemble spreads of

$\lambda_{PM2.5}^f$, $\lambda_{SO2}^f$, $\lambda_{NO}^f$ and $\lambda_{NH3}^f$ at a certain level. Several sensitive experiments were performed to investigate $\beta$ (1.2, 1.5, 1.8, 2, 2.5). It is seemed that reasonable results can be obtained when the ensemble spread of the emission scaling factors $\lambda_{PM2.5}^f$

ranged from 0.1 to 1. Finally, $\beta = 1.5$ was chosen in Peng et al. (2017). The area- averaged ensemble spreads of $\lambda_{PM2.5}^f$, $\lambda_{SO2}^f$, $\lambda_{NO}^f$ and $\lambda_{NH3}^f$ were stably distributed around 0.5, 1.0, 1.5 and 0.8 respectively over the three sub-regions: Beijing–Tianjin–

Hebei region, Yangtze River delta and Pearl River delta. It is apparent that the ensemble spread of $\lambda_{SO2}^f$ and $\lambda_{NO}^f$ is a little large due to the same $\beta$.

Therefore, it is better to choose different inflation factors for different emission scaling factors. We have performed several sensitive experiments to determine the value of $\beta$ over a 2-day period before the experiments written in the manuscript. The criterion we choose $\beta$ is to keep the ensemble spread of the scaling factors ranging from 0.1 to 1 in most model area. Finally, $\beta$ is chosen as 1.3, 1.4, 1.3, 1.2, 1.2, and 1.4

for $\lambda_{PM2.5}^f$, $\lambda_{PM10}^f$, $\lambda_{SO2}^f$, $\lambda_{NO}^f$, $\lambda_{NH3}^f$ and $\lambda_{CO}^f$ (See ReFig. 1)

Perhaps there are very few negative values for $(\kappa_{i,t})_{inf}$ after inflation. A quality control procedure is performed for $(\kappa_{i,t})_{inf}$ before further appliance. All these negative data were set as 0 in this work. Then $(\kappa_{i,t})_{inf}$ were re-centered to ensure the ensemble mean values of $(\kappa_{i,t})_{inf}$ were all 1. Then, another quality control procedure is performed for $\lambda_{i,t}^a$ to keep them positive. Thus, all $\lambda_{i,t}^f$ and $\lambda_{i,t}^a$ could be positive.

We have added these sentences in Line 158-163, 166-169, Page 6.

[Figure]

ReFig. 1. Spatial distribution of the ensemble spread for $\lambda^f_{PM2.5}$,$\lambda^f_{PM10}$,$\lambda^f_{SO2}$,

$\lambda^f_{NO}$,$\lambda^f_{NH3}$ and $\lambda^f_{CO}$ at the lowest model level at 0000 UTC 6 October 2014 in the

NCP region.

4) **L257-259: How did you perturb the initial conditions, lateral boundary**

**conditions and emissions? In other words, please provide how you estimated the**

**background uncertainty and spatial correlations (i.e. background covariance**

**structures) for the chemical state variables in adding perturbations?**

Before the first DA cycle, a 50-member ensemble of four-day spin-up forecasts was performed, with perturbed meteorological initial conditions (ICs), lateral boundary conditions (LBCs) and emissions, from 0000 UTC 1 October to 2300 UTC 4 October

2014. The perturbed meteorological ICs and LBCs are created by adding Gaussian random noise (Torn et al., 2006) to the temperature, water vapor, velocity, geopotential height and dry surface pressure fields of the products of the National Centers for

Environmental Prediction Global Forecast System (GFS) by WRFDA. The perturbed emissions were generated also by adding Gaussian random noise with a standard deviation of 10 percent of the corresponding anthropogenic emissions. The aerosol ICs were zero and the aerosol LBCs were idealized profiles embedded within the

WRF/Chem model. They are the same as in Peng et al. (2017). It is noted that the perturbed emissions were only used in the initial part.

In the DA part, the ICs were the analysis of the previous DA cycle, the meteorological LBCs were the perturbed LBCs. The anthropogenic emissions, $\mathbf{E}_{PM2.5}^f$,

$\mathbf{E}_{PM10}^f$, $\mathbf{E}_{SO2}^f$, $\mathbf{E}_{NO}^f$, $\mathbf{E}_{NH3}^f$, $\mathbf{E}_{CO}^f$, sulfate $\mathbf{E}_{SO4}^f$ and nitrate $\mathbf{E}_{NO3}^f$ are calculated by using the forecast emission scaling factors. Other species, such as the organic compounds $\mathbf{E}_{org}$ and elemental compounds $\mathbf{E}_{BC}$, are perturbed by adding Gaussian random noise. Since the emissions are calculated by EQ. (1), their background uncertainties and the spatial correlations are completely dependent on those of the corresponding emission factors. The forecast scaling factors are calculated by EQ. (2)

~ (5). And no other perturbations are added to the scaling factors; no other correlations are assumed for the scaling factors.

The experimental design is the same as in Peng et al (2017). We have rewritten briefly in Section 4 to avoid the repetition (Line 272-293, Page 10-11).

**5) L275-279 and Figure 2: This is very promising. I would imagine that the impacts**

**of other sources of uncertainties in air-quality forecast that were not directly**

**considered in this study (such as chemical schemes and parameterizations in**

**forecast model, and meteorology) were indirectly considered through the well-**

**calibrated inflations of state variables. Could you please make a comment about**

**the impacts of these other uncertainty sources in discussion section? I believe it**

**would be helpful for the future readers of this manuscript.**

It is true that the impacts of other sources of uncertainties in air-quality forecast (such as chemical schemes and parameterizations in forecast model, and meteorology)

were not directly considered through the well-calibrated inflations of state variables.

EnKF assimilation is influenced greatly by model errors and observation errors. But it is very difficult to accurately evaluate the uncertainties of models, though the covariance inflation technique was simply applied for all state variables to roughly compensate for model errors. Therefore, we can only obtain suboptimal results through

EnKF assimilation.

We have added the above paragraph in Lines 476-482, Page 17.

**6) Figure 4: It is not clear to me what "The shaded backgrounds indicate the distribution of the observations, where the top edge represented the 90th percentile and the bottom edge the 10th percentile" means. Does this distribution represent the observation values of individual sites in the Beijing–Tianjin–Hebei (BTH) region? Are other (red, black, pink, blue and light green) thick lines average of all sites in BTH region? The purpose to show these two values together is unclear to me, since the grey shaded line and other thick lines do not seem to be comparable each other. I would recommend to add more explanations about this figure, or to remove the grey shaded lines.**

Yes. the grey shaded line represent the distribution of the observation values of individual sites in the Beijing–Tianjin–Hebei (BTH) region. Other (red, black, pink, blue and light green) thick lines represent the average values of all sites in BTH region. No more information could be obtained from the grey shaded line since the average values of observations (red line) were shown. Thus we remove the grey shaded lined in Figure 4.

**List of specific comments:**

**1) L174: Please change "chose" to "chosen".**

We have revised the word in Line 188, Page 6.

**2) L296: I think "was able to" better fits with this context than "could".**

We have changed the word in Line 328, Page 12.

**3) Figure 4: The acronyms of "an" and "ct" is not described (although they can be guessed from the figure caption). Could you please add the explanation of those acronyms in the figure captions, such as "the analysis (referred to as "an", pink line)"?**

We have changed theese in Line 848-855, Page 32.

**4) Figure 11: Please add the explanation of grey shaded lines in the top panels.**

We remove the grey shaded lined in Figure 11, similar to Figure 4.

**Response to Reviewer #2's comments:**

We thank Referee # 2 for his thoughtful comments and suggestions that have helped to improve this manuscript. Our responses to comments (in bold style) and the corresponding changes to the manuscript are detailed below. In particular, we have added a simulation using the optimized emissions from 5 to 16 October 2014 according to his suggestions.

**There is not much to criticize about the manuscripts as it relies on the assimilation methodology previously described by Peng et al. (2017). (1) Since the assimilation experiment was conducted over a ten-day period it is uncertain if the conclusions about different performance of forecasts for various species would hold in a general. The most interesting are results on emission factors. (2) Did you encounter negative lambdas and if so what did you do about them? (3) An ultimate test of the optimized emissions would compare a simulation using the optimized emissions with a control. (4) Would an ENFK run with concentrations as state vectors using optimized emissions be identical to the EnKF run with concentrations and emission factors as the state vectors? (5) Link http://113.108.142.147:20035/emcpublish (p. 3) would be a valuable data source on pollution over China for many users but the access requires installation of Microsoft Silverlight a software for watching videos. That seems odd and is not be allowed on government computers. Could that be ameliorated?**

**(1) Since the assimilation experiment was conducted over a ten-day period it is uncertain if the conclusions about different performance of forecasts for various species would hold in a general.**

It is true that only a case was investigated in this work and it is uncertain if the conclusions about different performance of forecasts for various species would hold in a general. More case studies are needed to obtain general results in future works.

We have added the above paragraph in Lines 548-551, Page 19.

**(2) Did you encounter negative lambdas and if so what did you do about them?**

There are very few negative values for $(\kappa_{i,t})_{inf}$ after inflation (in Equation 3). A quality control procedure is performed for $(\kappa_{i,t})_{inf}$ before further appliance. All these negative data were set as 0 in this work. Then $(\kappa_{i,t})_{inf}$ were re-centered to ensure the ensemble mean values of $(\kappa_{i,t})_{\text{inf}}$ were all 1. Besides, another quality control procedure is performed for $\lambda_{i,t}^{\text{a}}$ to keep them positive. Thus, all $\lambda_{i,t}^{\text{f}}$ and $\lambda_{i,t}^{\text{a}}$ could be positive.

We have added the above paragraph in Lines 158-163, Page 6.

**(3) An ultimate test of the optimized emissions would compare a simulation using**

**the optimized emissions with a control.**

We have performed a simulation (fcEs) using the optimized emissions from 5 to

16 October 2014 to investigate the impact of optimized emissions on chemical simulations. Same as the control run, the ICs were the ensemble mean of the spin-up forecasts at 00:00 UTC on 5 October 2014. Thus the difference between the fcEs and the control run is the anthropogenic emissions. The results showed that the fcEs performed very similar to the control run in the whole in the BTH region (ReFig. 1). For

$PM_{2.5}$, $PM_{10}$ and CO, the values of the fcEs were a little smaller than those of the control run, which were consistent with the difference of the anthropogenic emissions. For $SO_2$

and $NO_2$, fcEs performed much better than the control run in most time though significant systematic overestimation still existed during the nighttime. For $O_3$, miner improvements were also gained due to the better simulation in fcEs for $NO_2$.

We have added the above paragraph in Line 443-453, Page 15. For ReFig.1, the cyan line (refer to as "fcEs") was added in Figure 4 to save space.

[Figure]

ReFig. 1. Time series of the hourly pollutant concentrations in the Beijing–Tianjin–Hebei (BTH) region obtained from observations (referred to as "obs", red line), the control run (referred to as "ct", black line), the analysis (referred to as "an", pink line), the simulation only using the optimized emissions (referred to as "fcEs", cyan line). The observations were obtained from the 47 independent sites in the BTH region. The modelled time series were interpolated to the 47 independent sites using the spatial bilinear interpolator method. Units: $\mu g\ m^{-3}$.

**(4) Would an EnFK run with concentrations as state vectors using optimized emissions be identical to the EnKF run with concentrations and emission factors as the state vectors?**

The optimized emissions are only the results of a mathematical optimum by utilizing observations. They are influenced greatly by model errors and observation errors. If the optimized emissions used in the EnFK experiment run with pure concentrations as state vectors are identical to the emissions assimilated in the joint

EnFK experiment run with concentrations and emission factors (representing emissions)

as state vectors, identical results may be obtained.

We have added the above paragraph in Line 116-121, Page 4-5.

**(5) Link http://113.108.142.147:20035/emcpublish (p. 3) would be a valuable data**

**source on pollution over China for many users but the access requires installation**

**of Microsoft Silverlight a software for watching videos. That seems odd and is not**

**be allowed on government computers. Could that be ameliorated?**

Yes, we agree with the reviewer that the requirement of installation of Microsoft

Silverlight software to view the data is odd. There is another website for the data:

http://www.resdc.cn/data.aspx?dataid=186. The data can be downloaded by request. If you are interested in the data, please contact the data manager of the website.

[revised manuscript text omitted]
_{i,t}^{a}$ to keep them positive. Thus, all $\lambda_{i,t}^{f}$ and $\lambda_{i,t}^{a}$ could be positive.

In this study, the ensemble forecast chemical fields of PM$_{25}$, PM$_{10}$, SO$_2$, NO, NH$_3$

and CO of the previous assimilation cycle are respectively used to calculate the emission scaling factors ($\lambda_{\text{PM2.5}}^{f}, \lambda_{\text{PM10}}^{f}, \lambda_{\text{SO2}}^{f}, \lambda_{\text{NO}}^{f}, \lambda_{\text{NH3}}^{f}, \lambda_{\text{CO}}^{f}$). Previous works (Peng et al., 2015, 2017) showed that reasonable results can be obtained when the ensemble spread of the emission scaling factors ranged from 0.1 to 1. In order to keep the ensemble spread of the scaling factors at this level in most model area, $\beta$ is chosen as 1.3, 1.4, 1.3, 1.2, 1.2, and 1.4 for the ensemble concentration ratios of P$_{25}$, P$_{10}$, SO$_2$,

NO, NH$_3$ and CO, respectively in Equation (3).

Then, the sources $\mathbf{E}_{i,t}^{f} = (\mathbf{E}_{\text{PM2.5}}^{f}, \mathbf{E}_{\text{PM10}}^{f}, \mathbf{E}_{\text{SO2}}^{f}, \mathbf{E}_{\text{NO}}^{f}, \mathbf{E}_{\text{NH3}}^{f}, \mathbf{E}_{\text{CO}}^{f})$ are calculated using equation (1).

From the perspective of PM$_{2.5}$ emissions, these include the unspeciated primary sources of PM$_{2.5}$ $\mathbf{E}_{PM2.5}$, sulfate $\mathbf{E}_{SO4}$, and nitrate $\mathbf{E}_{NO3}$. We updated $\mathbf{E}_{PM2.5}$, $\mathbf{E}_{SO4}$

and $\mathbf{E}_{NO3}$ (including the nuclei and accumulation modes) following Peng et al. (2017).

**2.3 DA algorithm**

The assimilation algorithm employed was the EnSRF proposed by Whitaker and Hamill (2002). The EnKF proposed by Evensen (1994) needs perturbations of observations in practice. Compared to the original EnKF, the EnSRF obviates the need to perturb the observations and avoids additional sampling errors introduced by perturbing observations.

We used the same EnSRF as in Schwartz et al. (2012, 2014). The ensemble member was chosen as 50. The localization radius was chosen as 607.5 km, so EnSRF

analysis increments were forced to zero at 607.5 km away from an observation (Gaspari and Cohn, 1999). The posterior (after assimilation) multiplicative inflation factor was chosen as 1.2 for all the concentration analysis.

**2.4 State variables**

The DA system provides joint analysis of ICs and emissions following Peng et al.

(2017). Among them, 16 WRF-Chem/GOCART aerosol variables are included as the state variables. Besides, chemical species, such as SO$_2$, NO$_2$ and O$_3$ are also included because they are the most important gas-phase precursors or oxidants of the secondary inorganic aerosols. CO is also assimilated because CO is an important tracer of combustion sources, as well as a precursor of O$_3$ beyond NO$_2$ (Parrish et al., 1991). The state variables of the emission scaling factors are $\boldsymbol{\lambda} =$

( $\boldsymbol{\lambda}_{PM2.5}, \boldsymbol{\lambda}_{PM10}, \boldsymbol{\lambda}_{SO2}, \boldsymbol{\lambda}_{NO}, \boldsymbol{\lambda}_{NH3}, \boldsymbol{\lambda}_{CO}$).

Similar to weak-coupling DA, the DA system simultaneously updates both the ICs and the emissions, but with no cross-variable update, in order to avoid the effects of spurious multivariate correlations in the background error covariance that may develop due to the limited ensemble size and errors in both the model and observations (Miyazaki et al. 2012).

For the PM2.5 observations, the observation operator is expressed as (Schwartz et al., 2012)

$$y_{pm25}^f = \rho_d[P_{25} + 1.375S + 1.8(OC_1 + OC_2) + BC_1 + BC_2$$

$$+D_1 + 0.286D_2 + S_1 + 0.942S_2],\qquad(6)$$

where $\rho_d$ is the dry air density; $P_{25}$ is the fine unspectiated aerosol contributions; S

represents sulfate; $OC_1$ and $OC_2$ are hydrophobic and hydrophilic organic carbon respectively; $BC_1$ and $BC_2$ are hydrophobic and hydrophilic black carbon respectively;

$D_1$ and $D_2$ are dusts with effective radii of 0.5 and 1.4 μm espectively; $S_1$ and $S_2$ are sea salts with effective radii of 0.3 and 1.0 μm espectively. In fact, PM2.5 observations were only used to analyze $P_{25}$, S, $OC_1$, $OC_2$ $BC_1$, $BC_2$, $D_1$, $D_2$, $S_1$, $S_2$ and $\lambda_{PM2.5}$. Since we had no NH3 observations, PM2.5 observations were also used to analyze $\lambda_{NH3}$ (see

Table 2). For other control variables, PM2.5 observations were not allowed to alter them.

For the PM10 observations, the PM10 observation operator is expressed as (Jiang et al., 2013)

$$y_{pm10}^f = \rho_d[P_{10} + P_{25} + 1.375S + 1.8(OC_1 + OC_2) + BC_1 + BC_2$$

$$+D_1 + 0.286D_2 + D_3 + 0.87D_4 + S_1 + 0.942S_2 + S_3].\ (7)$$

Thus,

$$y_{pm10-2.5}^f = \rho_d[P_{10} + D_3 + 0.87D_4 + S_3],\qquad(8)$$

meaning that, in the assimilation experiments, we did not use the PM10 observations directly. In equation (13) and (14), $P_{10}$ denotes the coarse-mode unspectiated aerosol contributions; $D_3$ and $D_4$ are dusts with effective radii of 2.4 and 4.5 μm respectively;

$S_3$ is sea salt with effective radii of 3.25 μm. We used the PM10-2.5 observations (the differences between the PM10 observations and the PM2.5 observations, $y_{pm10-2.5}^o =$

[revised manuscript text omitted]